# MindMix: A Multimodal Foundation Model for Auditory Perception Decoding via Deep Neural-Acoustic Alignment

**Rui Liu[1], Zhige Chen[1], Shu Peng[1], Wenlong You[1], Zhi-An Huang[2], Jibin Wu[1],\* Kay Chen Tan[1]**
[1]The Hong Kong Polytechnic University, [2]City University of Hong Kong (Dongguan)
`{ruiliu, shupeng, wenlong.you, jibin.wu, kctan}@polyu.edu.hk`,
`zhige.chen@connect.polyu.hk, huang.za@cityu.edu.cn`

## Abstract

Decoding complex auditory experiences from non-invasive EEG is a rapidly emerging field that holds significant promise for advancing both fundamental neuroscience and human-machine interaction technologies. Recent developments in EEG foundation models have yielded powerful neural representations that are promising for auditory decoding. However, the effectiveness of these models remains fundamentally constrained by their limited integration with acoustic stimulus information. Specifically, the lack of deep coupling between neural signals and auditory inputs hampers the models' ability to generalize effectively across diverse auditory tasks. To bridge this gap, we introduce MindMix, a multimodal foundation model designed to bridge the gap between unimodal EEG foundations and task-specific auditory decoders. MindMix employs a two-stage training strategy: first, a high-capacity EEG encoder is pre-trained on over 3,000 hours of EEG data to learn generalized EEG features that can transfer across tasks and subjects. Second, the model learns the neural-acoustic mapping using over 100 hours of paired data, facilitated by our novel Cross-Attention Low-Rank Alignment module, which facilitates fine-grained, cross-modal information integration. Experimental results demonstrate that MindMix substantially surpassing existing baselines across a range of auditory decoding tasks, including auditory attention decoding, auditory emotion recognition, and cross-modal retrieval. This work thus establishes a foundation for future research in multimodal brain decoding and auditory brain-computer interfaces. Our code is available at https://github.com/CookieMikeLiu/MindMix.

## 1 Introduction

Auditory perception plays a central role in how humans interact with the world, shaping language understanding, environmental awareness, and social communication Opoku-Baah et al. (2021). Decoding the brain's representation of auditory experiences is a core pursuit in cognitive neuroscience and a key capability for brain-computer interface (BCI) systems Mahrooz et al. (2024). Recent advances show that brain signals contain rich acoustic and semantic information, enabling the direct interpretation of internal auditory experiences from neural activity Li et al. (2023); Chen et al. (2024a); Mathis et al. (2024). Among available techniques, electroencephalography (EEG) is widely used for its non-invasiveness and high temporal resolution Défossez et al. (2023); Li et al. (2024); Liu et al. (2024). However, decoding rich, naturalistic auditory experiences is fundamentally hindered by EEG's inherent limitations: a low signal-to-noise ratio and high inter-subject variability Piastra et al. (2021); Bonetti et al. (2024); Oxenham & Kreft (2016).

Historically, these challenges were compounded by task-specific modeling strategies that showed poor generalization across tasks and subjects Crosse et al. (2016); Yan et al. (2024b;a). A recent paradigm shift towards EEG foundation models, such as EEGPT Wang et al. (2024b), LaBraM Jiang et al. (2024), and HEAR Chen et al. (2025), has begun to address this by learning transferable repre-

---

*Corresponding authors

sentations from massive unlabeled EEG datasets. However, their effectiveness in auditory decoding is fundamentally limited by their unimodal nature. When trained exclusively on EEG signals, their representations are not optimized to align with the underlying structure of acoustic information, as they lack exposure to corresponding auditory stimuli. This highlights a critical research gap: the lack of a unified framework capable of learning well-aligned multimodal representations for robust and versatile auditory neural decoding Poziomska et al. (2024); Liu et al. (2025).

To bridge this gap, we introduce MindMix, the first multimodal foundation model specifically designed to learn a deeply aligned neural-acoustic representation from large-scale, paired EEG-audio data. The design of MindMix directly addresses the challenges of cross-modal learning. Its architecture features two key innovations: (1) a high-capacity EEG encoder, trained from scratch with a multi-task objective to robustly capture complex neural dynamics from noisy signals, and (2) a novel Cross-Attention Low-Rank Alignment (CALRA) module, which enables fine-grained alignment between neural patterns and acoustic features. CALRA moves beyond simple projection-based alignment to facilitate deep interaction between modalities. The entire framework is optimized end-to-end via a contrastive learning objective on over 100 hours of paired data, which explicitly forces the model to map corresponding EEG-audio pairs to nearby points in a shared embedding space. Our main contributions are summaris ed as follows:

- We introduce MindMix, the first multimodal foundation model designed to learn fine-grained and deeply aligned neural-acoustic representations, enabling robust performance across diverse auditory decoding tasks.

- We propose CALRA, a novel neural architecture for cross-modal alignment that enables fine-grained and auditory-type-aware interaction between neural and acoustic modalities.

- Extensive experimental results on MindMix demonstrate superior cross-modal alignment, leading to significantly improved neural decoding performance across a range of auditory perception tasks, including auditory attention decoding, auditory emotion recognition, and cross-modal music retrieval.

## 2 RELATED WORK

### 2.1 AUDITORY PERCEPTION DECODING FROM BRAIN SIGNALS

Early work on auditory decoding focused on reconstructing speech features from brain signals using linear models like regression or temporal response functions O'Sullivan et al. (2015); Crosse et al. (2016); Ferrante et al. (2024); Dahan et al. (2025). While effective in controlled settings, these methods struggle with naturalistic scenarios due to their reliance on clean stimuli and long decision windows Mesgarani & Chang (2012). More recent deep learning models offer greater flexibility, with applications in auditory attention classification Su et al. (2022), speech/music discrimination Wang et al. (2024a); Niu et al. (2024), and affective state recognition Hu et al. (2024). However, these models remain predominantly task-specific. They are typically trained and evaluated in isolation, exhibit poor generalization across datasets or subjects Poziomska et al. (2024); Chen et al. (2024c), and fail to scale to diverse, real-world listening conditions.

### 2.2 EEG FOUNDATION MODELS FOR NEURAL REPRESENTATION LEARNING

To address the limitations of task-specific models, recent work has explored EEG foundation models, which learn general-purpose representations from large-scale datasets. Using Transformer-based architectures and self-supervised objectives, models like EEGPT Wang et al. (2024b), LaBraM Jiang et al. (2024), Neuro-GPT Cui et al. (2024) and CBraMod Wang et al. (2025) have achieved strong performance on clinical benchmarks such as epilepsy detection or sleep staging. However, a fundamental limitation of these models for auditory decoding is their lack of exposure to auditory stimuli. Pretrained exclusively on EEG signals, their representations are not optimized to align with acoustic structures, resulting in poor transferability to auditory decoding tasks. MindMix is explicitly designed to bridge this gap: by incorporating paired EEG-audio data during pretraining, it learns a shared embedding space that effectively aligns these modalities.

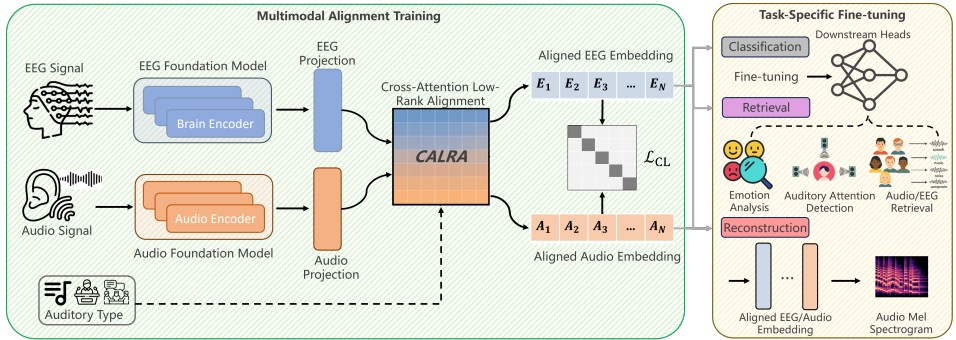

Figure 1: Overview of the proposed MindMix framework, which consists of an EEG encoder trained from scratch, a pretrained audio encoder, and our proposed CALRA module for fine-grained cross-modal alignment. Through large-scale pretraining with a contrastive objective, MindMix learns a unified EEG-audio representation space. This shared embedding facilitates strong generalization to a wide range of downstream auditory decoding tasks.

## 3 METHODOLOGY

### 3.1 OVERVIEW

We introduce MindMix, a multimodal foundation model that learns a unified embedding space to align EEG signals with corresponding auditory stimuli. As illustrated in Figure 1, given an input pair $(S_{\text{EEG}}, S_{\text{Audio}})$, MindMix uses a dual-stream architecture with two modality-specific encoders. These encoders produce feature embeddings $(E_{\text{proj}}, A_{\text{proj}})$, which are then processed by our core innovation, the CALRA module. CALRA performs deep interaction between the modalities, conditioned on the auditory type (e.g., speech, music), to produce the final aligned embeddings $(E_{\text{aligned}}, A_{\text{aligned}})$. The entire framework is optimized end-to-end via a contrastive learning objective, $\mathcal{L}_{\text{CL}}$ Chen et al. (2020), which maximizes the similarity between true $(E_{\text{aligned}}, A_{\text{aligned}})$ pairs while minimizing it for non-corresponding pairs within each training batch.

### 3.2 MODALITY-SPECIFIC ENCODERS

**EEG Encoder.** To address the core challenges of EEG signals, high inter-subject variability and heterogeneous channel configurations, the EEG encoder, $f_{\text{EEG}}$, is designed as a novel high-capacity architecture. As illustrated in Figure 2, this encoder is developed during the unimodal pre-training stage using a multi-task, self-supervised objective.

Our approach employs a channel-independent patching strategy to robustly handle heterogeneous electrode configurations. Given a raw signal $S_{EEG} \in \mathbb{R}^{C \times T}$ (where the channel count $C$ varies across datasets), we segment each channel independently into $K$ fixed-length temporal patches. These patches are passed through a temporal 1D convolution to obtain the initial embeddings $\tilde{X}$. Crucially, to learn discrete neural representations, we first quantize these initial embeddings $\tilde{X}$ into discrete neural tokens $v \in \mathcal{V}$ using a shared codebook. Following quantization, we construct the final input embedding $E_{patch}$ by adding learnable positional information to these tokens:

$$E_{\text{patch}} = v + \overline{\mathcal{T}} + \overline{\mathcal{E}} \tag{1}$$

where $\overline{\mathcal{T}}$ represents the learnable temporal embedding, which is added to each patch to indicate its relative temporal position index (1 to $K$) within the epoch; and $\overline{\mathcal{E}}$ represents the spatial (channel) embedding, implemented as a learnable lookup table that maps standard 10-20 system electrode identities (e.g., 'Cz', 'Pz') to unique vectors. This spatial embedding $\overline{\mathcal{E}}$ allows the model to distinguish the anatomical source of each patch regardless of the varying channel count $C$.

The main innovation of this stage is our unique pretraining methodology, which integrates two carefully designed self-supervised tasks. First, patch embeddings are quantized into discrete neural tokens $v \in \mathcal{V}$ using a shared codebook, which is optimized via a quantization loss $\mathcal{L}_{\mathcal{Q}}$. Subsequently, we compute two pretraining objectives:

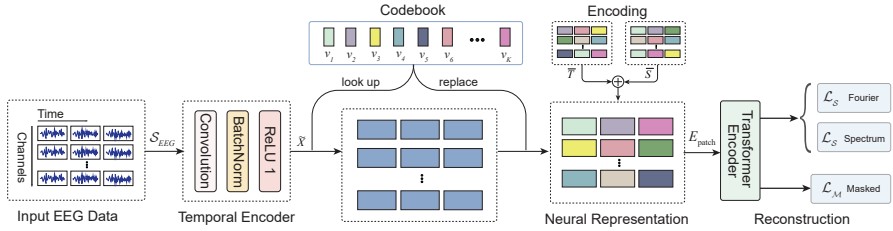

Figure 2: The multi-task pre-training architecture of the EEG encoder. The framework performs two tasks in parallel: one branch (top) reconstructs the Fourier spectrum from the full neural representations ($\mathcal{L}_\mathcal{S}$), while the main branch (bottom) performs masked token prediction ($\mathcal{L}_\mathcal{M}$) to learn robust features.

- **Masked Token Prediction**: A portion of the patch embeddings are randomly masked. A main Transformer encoder Vaswani et al. (2017) then predicts the original neural tokens of the masked patches from the visible ones, supervised by a masked modeling loss:

$$\mathcal{L}_\mathcal{M} = -\sum_{j \in \mathcal{M}} \log p(v_j | \tilde{X}_{\text{visible}}) \tag{2}$$

  where $\mathcal{M}$ is the set of masked patch indices.

- **Spectrum Reconstruction**: Concurrently, the unmasked patch embeddings are passed through a separate, smaller Transformer encoder. Its output reconstructs the Fourier spectrum (amplitude $A$ and phase $\psi$) of the original patches, supervised by a spectrum prediction loss:

$$\mathcal{L}_\mathcal{S} = \mathbb{E}_j \left[ ||\tilde{A}_j - A_j||^2 + ||\tilde{\psi}_j - \psi_j|| \right] \tag{3}$$

The total pre-training loss is a weighted sum of these objectives. The main Transformer from the masking task serves as the backbone for $f_{\text{EEG}}$. For multimodal alignment, we apply mean pooling over its output sequence and project it to produce the initial EEG embedding, $E_{\text{proj}}$.

**Audio Encoder.** Motivated by the strong performance of self-supervised pre-trained speech processing models Wang et al. (2021); Kunešová et al. (2024), we utilize the pretrained Wav2Vec 2.0 model Baevski et al. (2020) as our audio encoder, $f_{\text{Audio}}$. For each audio clip, we extract the final hidden state sequence from the Transformer, apply mean-pooling to obtain a single vector representation, and pass it through a linear projection layer to produce the initial audio embedding, $A_{\text{proj}}$.

### 3.3 Cross-Attention Low-Rank Alignment

The primary motivation for CALRA is to achieve a deep, robust semantic alignment capable of handling the unique challenges of auditory decoding. This task faces two specific hurdles: (1) the low signal-to-noise ratio and high non-linearity of EEG-audio mapping, for which standard "shallow projections" (like CLIP Radford et al. (2021)) are insufficient; and (2) the heterogeneity of stimuli (e.g., speech vs. music), where a uniform mapping fails to capture distinct neural response patterns. While "early fusion" methods (e.g., concatenation) could model these interactions, they break the dual-stream architecture required for efficient retrieval.

To bridge this gap, we propose CALRA, a global feature refinement module that implements a "refine-then-contrast" strategy. Instead of directly contrasting raw projections, CALRA injects deep, context-aware interactions into the embeddings before the loss calculation. Uniquely, it is designed to overcome the limitations of linear fusion by enforcing bilinear interactions in a shared bottleneck, allowing the model to capture fine-grained multiplicative dependencies that simple concatenation or co-attention Liu et al. (2023) cannot effectively model. The module consists of three synergistic components: a Type-specific Aligner to handle stimulus heterogeneity, a Bi-directional Cross-Attention mechanism for dynamic global context refinement, and a Shared Low-Rank Alignment to enforce deep bilinear fusion, which we detail below.

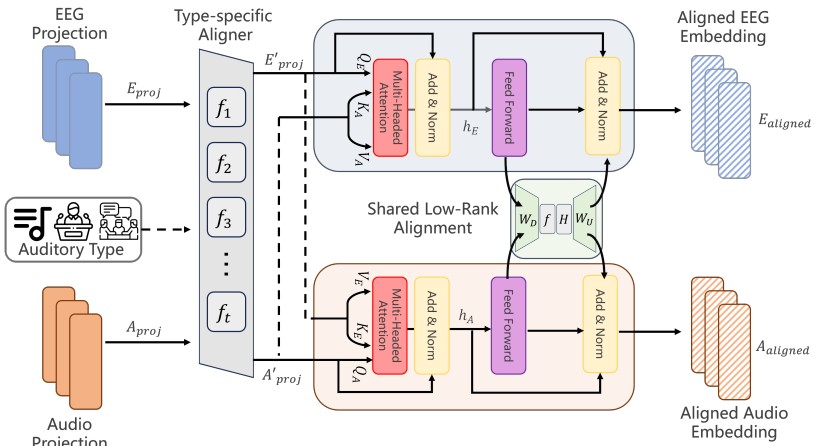

Figure 3: Overview of the proposed CALRA module. Given paired EEG and audio embeddings, CALRA performs auditory-type-specific alignment using multi-headed cross-attention and feed-forward networks. A shared low-rank bottleneck module further aligns the two modalities in a compact semantic space. The resulting aligned embeddings are used for contrastive pretraining, enabling robust cross-modal representation learning across diverse auditory conditions.

**Type-specific Aligner.** Given neural responses vary significantly for different types of auditory stimuli (e.g., speech vs. music), our aligner routes initial projections ($E_{\text{proj}}$, $A_{\text{proj}}$) through a learnable transformation $f_k$ corresponding to the auditory type label $k$:

$$(E'_{\text{proj}}, A'_{\text{proj}}) = f_k(E_{\text{proj}}, A_{\text{proj}}) \tag{4}$$

This allows the model to adopt optimal alignment strategies for different auditory stimulus types.

**Bi-directional Cross-Attention.** Following type-specific alignment, CALRA utilizes a bi-directional cross-attention mechanism to enable each modality to dynamically integrate complementary global context from the other. Unlike standard local token matching, we operate on the global projected embeddings to enforce holistic alignment. Given the projected global vectors $E'_{\text{proj}} \in \mathbb{R}^{1 \times D}$ and $A'_{\text{proj}} \in \mathbb{R}^{1 \times D}$ (where $D$ denotes the alignment dimension and the temporal dimension is aggregated via global pooling), this exchange occurs simultaneously:

- **Audio-to-EEG Alignment**: The EEG sequence ($Q_E$) retrieves relevant information from the audio sequence ($K_A, V_A$):

$$E'_{\text{interacted}} = \text{MultiHeadAttention}(Q_E, K_A, V_A) \tag{5}$$

- **EEG-to-Audio Alignment**: Symmetrically, the audio sequence ($Q_A$) retrieves neural features from the EEG sequence ($K_E, V_E$):

$$A'_{\text{interacted}} = \text{MultiHeadAttention}(Q_A, K_E, V_E) \tag{6}$$

Following standard practice in Transformer architecture Vaswani et al. (2017), residual connections and Layer Normalization are applied, yielding representations $h_E$ and $h_A$. These are then passed to our Shared Low-Rank Alignment module.

**Shared Low-Rank Alignment.** To enforce semantic consistency, we employ a shared low-rank fusion mechanism. Unlike standard CLIP Radford et al. (2021) which relies on a shallow linear dot-product, we aim to capture the complex, non-linear dependencies between neural and acoustic features. By projecting $h_E$ and $h_A$ into a shared bottleneck and fusing them via an element-wise product ($\odot$), this module enforces a bilinear interaction:

$$E_{\text{feedback}} = W_{D,eeg}\left(H_{\text{shared}}\left(W_{U,eeg}(h_E) \odot W_{U,audio}(h_A)\right)\right) \tag{7}$$

$$A_{\text{feedback}} = W_{D,audio}\left(H_{\text{shared}}\left(W_{U,eeg}(h_E) \odot W_{U,audio}(h_A)\right)\right) \tag{8}$$

where $W_{U,\cdot}, W_{D,\cdot}$ are modality-specific projection layers, and $H_{\text{shared}}$ is a shared non-linear layer. The final aligned embeddings are obtained by integrating this feedback via a residual connection:

$$E_{\text{aligned}}, A_{\text{aligned}} = \text{LayerNorm}(h_E + E_{\text{feedback}}), \text{LayerNorm}(h_A + A_{\text{feedback}}). \qquad (9)$$

We chose this low-rank structure because it efficiently approximates computationally expensive tensor fusion operations Liu et al. (2018); Yu et al. (2017), enabling the model to capture rich multiplicative feature interactions Fukui et al. (2016); Zadeh et al. (2017). This is theoretically superior to simple linear combinations for disentangling the intricate correlations between brain signals and auditory stimuli. It also need to note that, this multiplicative fusion architecture differs fundamentally from parameter-efficient strategies like LoRA Hu et al. (2022), which employ low-rank matrices for additive weight adaptation rather than for modeling the joint distribution of multimodal features.

Finally, we distinguish our design from recent works like MGCA Wang et al. (2022) and CARZero Lai et al. (2024), which primarily focus on fine-grained local token matching or modifying the similarity scoring function Zhang et al. (2025). In contrast, CALRA operates as a pre-loss refinement step for global representations. By enhancing the embeddings themselves ($E_{\text{aligned}}, A_{\text{aligned}}$) rather than altering the loss mechanism, it preserves the training stability of standard contrastive learning while capturing deep, context-aware dependencies that simpler projections miss.

### 3.4 PRE-TRAINING VIA CONTRASTIVE ALIGNMENT

We optimize the MindMix framework using a contrastive learning objective, inspired by CLIP Radford et al. (2021), on the final aligned embeddings. The goal is to maximize the cosine similarity of true EEG-audio pairs while minimizing it for incorrect pairs within a mini-batch. This is framed as a directed prediction problem where, for each EEG embedding, the model must identify its correct audio counterpart from all available options. This EEG-to-audio direction directly mirrors our downstream decoding tasks and avoids the potential instability of aligning a single audio stimulus with its many possible neural responses. We use the InfoNCE loss, which is equivalent to a cross-entropy loss over the similarity scores, averaged over all samples in the batch:

$$\mathcal{L}_{\text{CL}} = -\frac{1}{N} \sum_{i=1}^{N} \log \frac{\exp(\text{sim}(E_{aligned,i}, A_{aligned,i})/\tau)}{\sum_{j=1}^{N} \exp(\text{sim}(E_{aligned,i}, A_{aligned,j})/\tau)} \qquad (10)$$

where $\text{sim}(u, v)$ is the cosine similarity and $\tau$ is a learnable temperature. Minimizing this objective jointly trains the entire framework ($f_{\text{EEG}}$, $f_{\text{Audio}}$, and CALRA), forcing the model to learn a semantically rich embedding space where neural activity is meaningfully aligned with auditory content.

## 4 EXPERIMENTS

### 4.1 EXPERIMENTAL SETUP

**Datasets and Tasks.** Our experiments follow a three-stage pipeline: unimodal pre-training, multimodal alignment, and downstream task fine-tuning. For pre-training, we leverage large-scale public corpora, including over 3,500 hours of general EEG data and over 100 hours of paired EEG-audio data (summarized in Table 1 and detailed in Appendix A.1). We evaluate MindMix's generalization capabilities on a diverse set of downstream auditory decoding tasks, including Auditory Attention Decoding (on KUL, DTU, and ESAA), Emotion Analysis (on PME4 and HR-EEG4EMO), and Music Retrieval (on MAD-EEG). To ensure a fair evaluation, all downstream task datasets were held out from unimodal pre-training and multimodal alignment stages. Standardized data preprocessing protocols are detailed in Appendix A.2. For reproducibility, specific implementation details and the full hyperparameter configurations for the EEG encoder, CALRA module, and optimization process as well as our rigorous negative sampling policy (in Appendix A.3).

**Evaluation Protocol** To ensure a fair and rigorous comparison with existing SOTA methods, we adopt the widely established standard evaluation protocols. For all downstream tasks, we conduct experiments using a subject-specific (within-subject) protocol, implemented via a strict 5-fold cross-validation scheme. In this setup, the data for each subject are randomly partitioned into 5 folds, using a 70%/10%/20% split for training, validation and testing within each fold. We report the

Table 1: Overview of all datasets used across three training stages in our study.

| Stage | Dataset | Hours | Channels | Modality | Paradigm |
|---|---|---|---|---|---|
| *Stage 1: Unimodal Pre-training* | | | | | |
| | BCI-IV-2A Tangermann et al. (2012) | 13.4h | 22 | EEG Signal | Motor Imagery |
| | HGD Schirrmeister et al. (2017) | 28.7h | 133 | EEG Signal | Motor Imagery |
| | OpenBMI Lee et al. (2019) | 91.6h | 62 | EEG Signal | Motor Imagery |
| | EEGMat Zyma et al. (2019) | 2.4h | 20 | EEG Signal | Workload Analysis |
| | TUEP Veloso et al. (2017) | 631.8h | 14 | EEG Signal | Epilepsy Detection |
| | TUEV Veloso et al. (2017) | 148.7h | 8 | EEG Signal | Event Classification |
| | HMCSleep Alvarez (2021) | 582.5h | 8 | EEG Signal | Sleep Detection |
| | CAPSleep Terzano et al. (2001) | 1004.5h | 20 | EEG Signal | Sleep Detection |
| | CMBMIT Shoeb (2009) | 1060.9h | 8 | EEG Signal | Sleep Detection |
| | **Total Hours:** | **3564.5h** | | | |
| *Stage 2: Multimodal Alignment Training* | | | | | |
| | ds004356 Singh et al. (2024) | 38.9h | 34 | EEG + Audio | Music/Speech Listening |
| | zenodo_4518754 Mundanad et al. (2021) | 11.6h | 255 | EEG + Audio | Speech AAD |
| | zenodo_10260082 Thornton et al. (2023) | 12.0h | 2 | EEG + Audio | Speech AAD |
| | Brennan_2018 Brennan & Hale (2019) | 10.1h | 61 | EEG + Audio | Story Listening |
| | Broderick_2018 Broderick et al. (2018) | 19.1h | 128 | EEG + Audio | Story Listening |
| | Le_Petit_Prince Momenian et al. (2024) | 17.3h | 64 | EEG + Audio | Story Listening |
| | **Total Hours:** | **109.0h** | | | |
| *Stage 3: Downstream Task Fine-tuning* | | | | | |
| | MAD-EEG Cantisani et al. (2019) | 4.2h | 20 | EEG + Audio | Music Retrieval |
| | KUL Das et al. (2016) | 19.2h | 64 | EEG + Audio | Speech AAD |
| | DTU Fuglsang et al. (2017) | 15.0h | 64 | EEG + Audio | Speech AAD |
| | ESAA Li et al. (2022) | 12.7h | 64 | EEG + Audio | Speech AAD |
| | PME4 Chen et al. (2022) | 4.6h | 8 | EEG + Audio | Emotion Analysis |
| | HR-EEG4EMO Cantisani et al. (2019) | 10.0h | 128 | EEG + Audio | Emotion Analysis |

mean and standard deviation across these 5 folds. Crucially, all reported results utilize raw window-level metrics (accuracy per 2-second segment) rather than aggregated trial-level scores, providing a conservative and fine-grained assessment of decoding performance. Evaluation metrics include Balanced Accuracy and Weighted F1-score for AAD and emotion analysis, and standard Duo/Trio Accuracy for music retrieval. The detailed evaluation metrics can be found in Appendix A.4.

However, specifically regarding the Speech AAD task, we acknowledge that the mainstream within-subject splitting may introduce potential data leakage risks due to temporal correlations, as highlighted by Puffay et al. (2023). To address this, we additionally introduce a rigorous between-trial evaluation protocol, where training and testing segments are strictly drawn from disjoint trials (e.g., different stories or sessions) to prevent temporal overlap and artifact leakage. The detailed results of this robust evaluation are provided in Appendix A.5.

## 4.2 DOWNSTREAM EXPERIMENT RESULTS

MindMix was evaluated against strong baselines, including task-specific SOTA models (e.g., DBP-Net, AADNet) and powerful unimodal EEG foundation models (e.g., LaBraM, EEGPT). The detailed information about the compared baseline is provided in Appendix A.6. As shown in Table 2, MindMix substantially outperforms all baselines across all downstream tasks. It achieves near-perfect performance in Speech AAD (e.g., 99.82% on KUL) and establishes a new SOTA in other tasks with large margins (e.g., over 10 percentage points on PME4), underscoring the effectiveness of our multimodal strategy. A deeper analysis of these results reveals two critical findings. **First**, the unimodal EEG foundation models, such as LaBraM and CBraMod, consistently underperform when compared to task-specific SOTA models like DBPNet and DARNet. For instance, on the KUL dataset, LaBraM and CBraMod achieve accuracies of only 63.30% and 68.42%, respectively, falling far short of the 94.81% achieved by DARNet. This exposes a key limitation of current foundation models: they are predominantly pre-trained on non-auditory tasks, rendering their generic representations suboptimal for decoding auditory perception. Furthermore, these large models are often highly sensitive to the data format and preprocessing pipelines; any mismatch with their original training configuration can lead to high performance variance and poor fine-tuning results (as empirically quantified in Appendix A.2).

**Second**, and more importantly, our results highlight a crucial distinction in the effectiveness of multimodal integration. For other task-specific multimodal models like MusicAAD (94.87% on KUL) and AADNet (93.18% on KUL), the performance improvement over their strong unimodal counterparts is relatively modest. In contrast, the performance leap demonstrated by MindMix to

Table 2: Performance comparison of MindMix against SOTA baselines on various downstream tasks and datasets. The best values are highlighted in bold, and the second-best values in each block are underlined. Based on the paired t-test with p-value correction ($\alpha = 0.05$), the $*$ indicates the marked method is significantly better than the compared methods.

| Task | Speech AAD | | | | | |
|---|---|---|---|---|---|---|
| **Method** | **KUL** | | **DTU** | | **ESAA** | |
| | Balanced Acc. | Weighted F1 | Balanced Acc. | Weighted F1 | Balanced Acc. | Weighted F1 |
| EEGNet Lawhern et al. (2018) | $0.7514 \pm 0.097$ | $0.7510 \pm 0.097$ | $0.6112 \pm 0.042$ | $0.6578 \pm 0.058$ | $0.7742 \pm 0.132$ | $0.7915 \pm 0.111$ |
| DBPNet Ni et al. (2024) | $0.9357 \pm 0.042$ | $0.9588 \pm 0.038$ | $0.8251 \pm 0.061$ | $0.8579 \pm 0.056$ | $0.8418 \pm 0.121$ | $0.7990 \pm 0.163$ |
| DARNet Yan et al. (2024b) | $\underline{0.9481 \pm 0.036}$ | $\underline{0.9567 \pm 0.025}$ | $0.8391 \pm 0.048$ | $0.8687 \pm 0.036$ | $\underline{0.9089 \pm 0.054}$ | $\underline{0.9389 \pm 0.042}$ |
| MusicAAD Niu et al. (2024) | $0.9318 \pm 0.018$ | $0.9487 \pm 0.016$ | $\underline{0.8456 \pm 0.038}$ | $\underline{0.8874 \pm 0.032}$ | $0.8343 \pm 0.042$ | $0.8442 \pm 0.038$ |
| AADNet Nguyen et al. (2025) | $0.7258 \pm 0.057$ | $0.7585 \pm 0.051$ | $0.6875 \pm 0.057$ | $0.7312 \pm 0.055$ | $0.8237 \pm 0.061$ | $0.8165 \pm 0.058$ |
| BENDR Kostas et al. (2021) | $0.5879 \pm 0.045$ | $0.6033 \pm 0.040$ | $0.5886 \pm 0.032$ | $0.6289 \pm 0.040$ | $0.7983 \pm 0.078$ | $0.8245 \pm 0.069$ |
| BIOT Yang et al. (2023) | $0.5538 \pm 0.051$ | $0.5941 \pm 0.047$ | $0.6233 \pm 0.029$ | $0.6588 \pm 0.032$ | $0.7588 \pm 0.077$ | $0.7895 \pm 0.072$ |
| EEGPT Wang et al. (2024b) | $0.6642 \pm 0.042$ | $0.7083 \pm 0.045$ | $0.6538 \pm 0.038$ | $0.6789 \pm 0.042$ | $0.8385 \pm 0.065$ | $0.8688 \pm 0.060$ |
| LaBraM Jiang et al. (2024) | $0.6330 \pm 0.052$ | $0.6498 \pm 0.043$ | $\underline{0.6582 \pm 0.046}$ | $0.6821 \pm 0.048$ | $\underline{0.8568 \pm 0.070}$ | $\underline{0.8544 \pm 0.071}$ |
| CBraMod Wang et al. (2025) | $\underline{0.6842 \pm 0.038}$ | $\underline{0.7252 \pm 0.041}$ | $0.6478 \pm 0.052$ | $0.6701 \pm 0.046$ | $0.8423 \pm 0.067$ | $0.8325 \pm 0.075$ |
| **MindMix (Ours)** | $\mathbf{0.9982 \pm 0.008^*}$ | $\mathbf{0.9991 \pm 0.004^*}$ | $\mathbf{0.9993 \pm 0.009^*}$ | $\mathbf{0.9996 \pm 0.005^*}$ | $\mathbf{1.0000 \pm 0.000^*}$ | $\mathbf{1.0000 \pm 0.000^*}$ |

| Task | Emotion Analysis | | | | Music Retrieval | |
|---|---|---|---|---|---|---|
| **Method** | **PME4** | | **HR-EEG4EMO** | | **MAD-EEG** | |
| | Balanced Acc. | Weighted F1 | Balanced Acc. | Weighted F1 | Duo Acc. | Trio Acc. |
| EEGNet Lawhern et al. (2018) | $0.5029 \pm 0.035$ | $0.4920 \pm 0.046$ | $0.6981 \pm 0.111$ | $0.7681 \pm 0.071$ | $0.5831 \pm 0.025$ | $0.4521 \pm 0.037$ |
| DBPNet Ni et al. (2024) | $0.5717 \pm 0.032$ | $0.5321 \pm 0.053$ | $\underline{0.8274 \pm 0.073}$ | $\underline{0.8458 \pm 0.064}$ | $0.7849 \pm 0.091$ | $0.7152 \pm 0.078$ |
| DARNet Yan et al. (2024b) | $0.5725 \pm 0.025$ | $0.5425 \pm 0.061$ | $0.8052 \pm 0.081$ | $0.8178 \pm 0.077$ | $0.7544 \pm 0.080$ | $0.7185 \pm 0.082$ |
| MusicAAD Niu et al. (2024) | $\underline{0.6142 \pm 0.062}$ | $\underline{0.6345 \pm 0.075}$ | $0.7648 \pm 0.084$ | $0.7852 \pm 0.069$ | $\underline{0.9425 \pm 0.028}$ | $\underline{0.8722 \pm 0.038}$ |
| AADNet Nguyen et al. (2025) | $0.6011 \pm 0.077$ | $0.5986 \pm 0.065$ | $0.7544 \pm 0.059$ | $0.7832 \pm 0.054$ | $0.8824 \pm 0.071$ | $0.8916 \pm 0.065$ |
| BENDR Kostas et al. (2021) | $0.5433 \pm 0.065$ | $0.5218 \pm 0.059$ | $0.6458 \pm 0.015$ | $0.6855 \pm 0.017$ | $0.6235 \pm 0.048$ | $0.6498 \pm 0.045$ |
| BIOT Yang et al. (2023) | $0.5224 \pm 0.071$ | $0.5359 \pm 0.069$ | $0.6352 \pm 0.023$ | $0.6487 \pm 0.019$ | $0.6485 \pm 0.052$ | $0.6798 \pm 0.049$ |
| EEGPT Wang et al. (2024b) | $0.5566 \pm 0.058$ | $0.5478 \pm 0.061$ | $0.7129 \pm 0.072$ | $0.7698 \pm 0.077$ | $0.7887 \pm 0.065$ | $0.7582 \pm 0.068$ |
| LaBraM Jiang et al. (2024) | $0.5868 \pm 0.056$ | $\underline{0.5936 \pm 0.052}$ | $\underline{0.7295 \pm 0.082}$ | $\underline{0.7829 \pm 0.081}$ | $0.7582 \pm 0.082$ | $0.7229 \pm 0.078$ |
| CBraMod Wang et al. (2025) | $\underline{0.6052 \pm 0.072}$ | $0.5841 \pm 0.088$ | $0.7285 \pm 0.078$ | $0.7748 \pm 0.074$ | $\underline{0.8011 \pm 0.069}$ | $0.7654 \pm 0.087$ |
| **MindMix (Ours)** | $\mathbf{0.7256 \pm 0.123^*}$ | $\mathbf{0.7089 \pm 0.135^*}$ | $\mathbf{0.8878 \pm 0.045^*}$ | $\mathbf{0.8869 \pm 0.046^*}$ | $\mathbf{0.9475 \pm 0.025^*}$ | $\mathbf{0.8824 \pm 0.042^*}$ |

99.82% is dramatic, which directly validates our central hypothesis: a deep cross-modal alignment is paramount, and simply combining modalities is not enough. This substantial lead in performance underscores the efficacy of MindMix's architectural design, which integrates powerful modality-specific encoders with an effective alignment strategy. Our framework enables deep interaction to capture the fine-grained relationship between brain activity and complex audio signals, which is crucial for robust neural decoding.

## 4.3 ABLATION STUDY AND ANALYSIS

To validate our architectural choices and quantify the contribution of each component, we perform comprehensive ablation studies, the results of which are summarized in Table 3.

**Effectiveness of the CALRA Module.** We first investigate the importance of our core contribution, CALRA, by comparing it against simpler alignment strategies. As shown in Table 3, replacing CALRA with a standard co-attention block or reverting to a simple CLIP-style projection ('w/o Alignment') leads to substantial performance degradation. Besides, to rigorously validate the structural advantage of our bilinear fusion, we further compare against a "Standard Concatenation-based Fusion (Concat-MLP)" baseline, which is the dominant strategy for vector-level integration. CALRA consistently outperforms this strong baseline (e.g., 0.8878 vs. 0.8574 on EEG4EMO). This empirical evidence confirms that the multiplicative interaction within CALRA captures complex cross-modal dependencies that simple concatenation cannot effectively model.

While Table 2 reports standard metrics for fair comparison, we further validated our model's robustness using the strict between-trial protocol defined in Section 4.1. As detailed in Appendix A.5, MindMix maintains a substantial and leading performance advantage under this challenging setting, although the absolute accuracy is lower as expected. This confirms that the model's superiority stems from genuine neuro-acoustic alignment rather than the exploitation of trial-specific artifacts.

**Impact of Modality Encoders.** To validate our encoder choice, we substituted our specialized EEG encoder with several alternatives. Both SOTA foundation models (LaBraM and CBraMod) and the classic EEGNet resulted in a significant performance drop, confirming the advantage of our custom pre-training strategy. Specifically, even when adapting the strong CBraMod backbone with our alignment module, the performance (96.37% on KUL) still falls short of our full MindMix

Table 3: Ablation studies on the main components of MindMix. All experiments are evaluated on two downstream tasks: emotion recognition on HR-EEG4EMO and auditory attention decoding on KUL. **Note:** For the "w/ LaBraM" and "w/ CBraMod" entries, we initialized the EEG encoder using their official pretrained weights and subjected them to our full multimodal alignment training before fine-tuning, ensuring a rigorous comparison of backbone capabilities.

| Model Configuration | EEG Encoder | Audio Encoder | Alignment Module | Emotion Acc. | AAD Acc. |
|---|---|---|---|---|---|
| **MindMix (Full Model)** | **Ours** | **Wav2Vec 2.0** | **CALRA (Ours)** | $0.8878 \pm 0.045$ | $0.9982 \pm 0.008$ |
| *Ablation on Alignment* | | | | | |
| w/ Co-Attention | Ours | Wav2Vec 2.0 | Co-Attention | $0.8629 \pm 0.053$ | $0.9785 \pm 0.021$ |
| w/ Concat-MLP | Our | Wav2Vec 2.0 | Concat-MLP | $0.8574 \pm 0.035$ | $0.9593 \pm 0.017$ |
| w/o Alignment | Ours | Wav2Vec 2.0 | Standard CLIP | $0.8483 \pm 0.038$ | $0.9535 \pm 0.015$ |
| *Ablation on EEG Encoder* | | | | | |
| w/ LaBraM | LaBraM | Wav2Vec 2.0 | CALRA (Ours) | $0.8588 \pm 0.041$ | $0.9744 \pm 0.012$ |
| w/ EEGNet | EEGNet | Wav2Vec 2.0 | CALRA (Ours) | $0.8555 \pm 0.047$ | $0.9442 \pm 0.011$ |
| w/ CBraMod | CBraMod | Wav2Vec 2.0 | CALRA (Ours) | $0.8642 \pm 0.039$ | $0.9637 \pm 0.010$ |
| *Ablation on Audio Encoder* | | | | | |
| w/ HuBERT | Ours | HuBERT | CALRA (Ours) | $0.8687 \pm 0.037$ | $0.9883 \pm 0.010$ |
| w/ Mel-spectrogram | Ours | Mel-spectrogram | CALRA (Ours) | $0.8432 \pm 0.035$ | $0.9448 \pm 0.015$ |
| *Dissection of the CALRA* | | | | | |
| w/o Type-specific Aligner | Ours | Wav2Vec 2.0 | CALRA (Ours) | $0.8675 \pm 0.035$ | $0.9853 \pm 0.010$ |
| w/o Shared Low-Rank | Ours | Wav2Vec 2.0 | CALRA (Ours) | $0.8557 \pm 0.040$ | $0.9742 \pm 0.012$ |
| w/o Cross-Attention | Ours | Wav2Vec 2.0 | CALRA (Ours) | $0.8482 \pm 0.036$ | $0.9435 \pm 0.013$ |

model (99.82%). For the audio stream, substituting the powerful Wav2Vec 2.0 with traditional Mel-spectrogram features causes a steep decline of up to 5.45% in AAD accuracy. This highlights that rich, pre-trained representations are essential for both the neural and acoustic modalities.

**Dissection of CALRA's Components.** Finally, we dissect the CALRA module to quantify the contribution of its three key innovations. The bi-directional cross-attention mechanism proves to be the most critical element, as its removal ('w/o Cross-Attention') causes the largest performance drop (up to 5.58% in AAD). The shared low-rank alignment also provides a vital contribution, with its removal ('w/o Shared Low-Rank') leading to a significant drop. The w/o Type-specific Aligner ablation, which simulates the absence of auditory type information at test time, causes only a minor performance drop. This indicates that while the type-specific routing is beneficial, our model does not critically rely on it and remains highly effective even when stimulus type is unknown. Together, these results confirm that all three components of CALRA are integral to its success, working synergistically to achieve a superior cross-modal alignment.

Finally, we also investigated the trade-off between decoding accuracy and temporal resolution (window size sensitivity), these additional results and detailed analyses are provided in Appendix A.7.

## 4.4 QUANTIFYING THE SYNERGY OF MULTIMODAL ALIGNMENT

To isolate and quantify the benefit of our multimodal approach, we conduct a critical analysis comparing the full MindMix model against its EEG-only counterpart. The results, presented in Figure 4, are striking and reveal a deep synergy fostered by the alignment of brain signals and audio. Notably, even this EEG-only counterpart is highly competitive on its own, demonstrating performance comparable to the SOTA unimodal baselines reported in Table 2. For instance, it outperforms the LaBraM baseline on the ESAA and MAD-EEG tasks. Furthermore, to validate its broader generalization capabilities, we benchmarked the encoder on standard non-auditory tasks (TUAB and BCIC-IV-2B). As detailed in Appendix A.8, our model achieves top-tier performance (e.g., ranking 1st on BCIC-IV-2B among foundation models), confirming its robustness as a general-purpose EEG encoder. This demonstrates that MindMix's success stems not just from learning robust EEG representations but from learning the relationship between the neural signal and the auditory stimulus.

## 4.5 NEUROSCIENTIFIC INTERPRETATION OF CROSS-MODAL ALIGNMENT

To provide a comprehensive assessment of the learned representations and validate their biological plausibility, we employ the "Stimulus Reconstruction" method Mesgarani & Chang (2012), a foundational approach in auditory neuroscience to quantify neural encoding. We adopt the pseudo-reconstruction framework from Défossez et al. (2023) to reconstruct audio Mel spectrograms from the aligned EEG embeddings ($E_{\text{aligned}}$), analyzing the results both qualitatively and quantitatively.

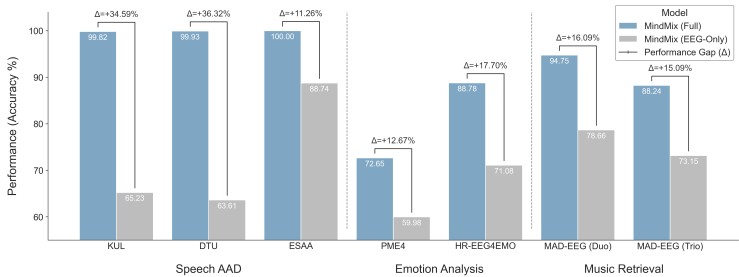

Figure 4: This figure compares the performance of the full MindMix model with its unimodal (EEG-Only) counterpart to isolate the performance gain from our cross-modal alignment strategy.

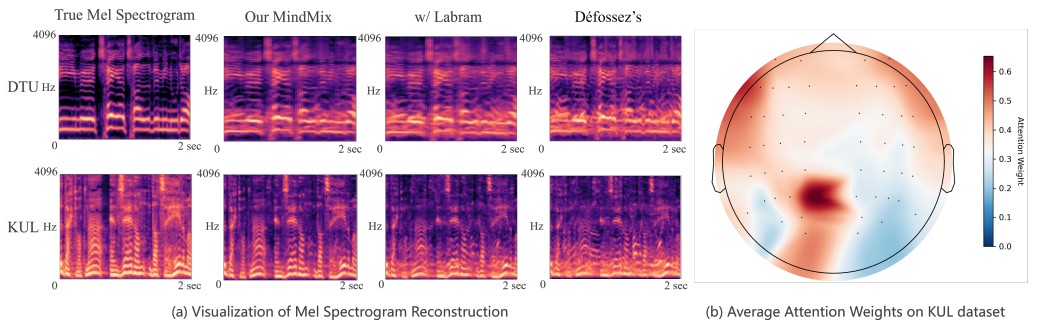

(a) Visualization of Mel Spectrogram Reconstruction      (b) Average Attention Weights on KUL dataset

Figure 5: Neuroscientific Interpretability Analysis. (a) Mel Spectrogram Reconstruction. (b) Spatial Attention Topography.

As illustrated in Figure 5(a), MindMix reconstructions faithfully capture fine-grained harmonic structures, whereas these details are blurred in the LaBraM variant and the baseline method. Quantitatively, MindMix achieves PCC scores of 0.88 (DTU) and 0.91 (KUL), substantially outperforming baselines (e.g., 0.67 and 0.61). Crucially, this high fidelity provides direct evidence that MindMix successfully encodes the spectro-temporal receptive fields of the auditory cortex, mapping neural activity accurately back to acoustic features.

To further investigate the physiological basis of these reconstructions, we visualized the spatial attention weights of the EEG encoder. As shown in Figure 5(b), the model exhibits a distinct, high-intensity activation cluster in the left temporal region. This distribution is neuroscientifically significant: it corresponds precisely to the primary auditory cortex and aligns with the well-established left-hemisphere lateralization for speech processing Défossez et al. (2023). The absence of high weights in the frontal pole further confirms that the model prioritizes genuine neural signatures over ocular artifacts.

## 5 CONCLUSION

This paper presents the first large-scale investigation into multimodal auditory brain decoding using paired EEG and audio data. We introduce MindMix, a novel foundation model featuring our CALRA module, which enables deep alignment between neural signals and sound. Our extensive experiments demonstrate that MindMix consistently and significantly outperforms SOTA baselines across a diverse set of downstream tasks, establishing a new and robust benchmark for the field. By successfully learning generalizable representations, this work significantly advances the capabilities of non-invasive BCIs and lays a critical foundation for understanding the interplay between neural and audio signals. While our results highlight the immense potential of this approach, we also underscore that the current scarcity of large-scale paired EEG-audio corpora is a primary bottleneck for the field, precluding a full investigation into the scaling laws of such foundation models. Future research will focus on scaling the MindMix framework to leverage increasingly larger datasets, with the goal of further advancing this field.

## ACKNOWLEDGEMENTS

This work was partially supported by the National Natural Science Foundation of China (Grant No. 62306259 and 62572413), the Research Grants Council of the Hong Kong SAR (Grant No. C5052-23G, PolyU25216423, PolyU15217424, and SRFS2526-5S04), The Hong Kong Polytechnic University (P0058445).

## ETHIC STATEMENT

This work adheres to the ICLR Code of Ethics. In this study, no human subjects or animal expreci-sions were included. All data sets used were sourced in accordance with relevant usage guidelines, to ensure no privacy violation. We have taken care not to achieve bias or discriminatory results in our research process. No personally identifiable information was used and no experiments were conducted that could raise privacy or security concerns. We are committed to maintaining transparency and integrity throughout the research process.

## REPRODUCIBILITY STATEMENT

We have made every effort to ensure that the results presented in this paper are reproducible. All code and model have been made publicly available in an anonymous repository to facilitate replication and verification, the experimental setup, including training steps, model configurations, and hardware details, is described in detail in the paper, We have also provided a full description of implementation description, to assist others in reproducing our experiments.

Additionally, all EEG datasets used in the paper, such as KUL, DUT, and ESAA etc, are publicly available, ensuring consistent and reproducible evaluation results.

We believe these measures will allow other researchers to reproduce our work and further advance the field.

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

## A   APPENDIX

### A.1 DATASET DESCRIPTION

**Stage 1: Unimodal Pre-training Corpus.**   For the initial unimodal pre-training stage, designed to build a robust EEG encoder, we assembled a large-scale and diverse corpus totaling over 3,500 hours of EEG data from nine public datasets. These datasets span a wide range of BCI paradigms and clinical applications, ensuring that the learned representations are generalizable and not biased towards a specific task. The collection includes:

- **BCI-IV-2A** Tangermann et al. (2012) and **HGD** Schirrmeister et al. (2017): Widely used motor imagery datasets.
- **OpenBMI** Lee et al. (2019): A large-scale dataset for various BCI paradigms.
- **EEGMat** Zyma et al. (2019): A benchmark for mental workload analysis.
- **TUEP** and **TUEV** Veloso et al. (2017): The Temple University Hospital EEG Corpus, containing extensive clinical data for epilepsy and event detection.
- **HMCSleep** Alvarez (2021), **CAPSleep** Terzano et al. (2001), and **CMBMIT** Shoeb (2009): Large public corpora for sleep stage analysis.

The unimodal pre-training of our EEG encoder utilizes over 3,500 hours of data, establishing a substantial foundation in terms of scale. For context, this exceeds the pre-training corpus of several notable EEG foundation models, such as LaBraM ( 2,500 hours) and EEGPT ( 200 hours), underscoring the large-scale nature of our unimodal representation learning.

**Stage 2: Multimodal Alignment Corpus.**   In the second stage, we focused on learning the crucial alignment between neural and acoustic representations. For this, we curated a multimodal corpus of over 100 hours of paired EEG and audio data from seven distinct datasets. This corpus features diverse auditory stimuli, including music, attended speech, and naturalistic story listening. The datasets include:

- **ds004356** Shan et al. (2024) and **zenodo_4518754** Mundanad et al. (2021): Publicly available datasets featuring subjects listening to music or competing speech streams in AAD paradigms.
- **zenodo_10260082** Thornton et al. (2023): An additional EEG-audio dataset for speech AAD tasks.
- **Brennan 2018** Brennan & Hale (2019), **Broderick 2018** Broderick et al. (2018), and **Le Petit Prince** Momenian et al. (2024): Datasets containing EEG recordings of subjects listening to naturalistic stories, providing rich, continuous audio stimuli.

**Stage 3: Downstream Task Datasets.** We evaluate our model on three distinct task families:

- **Auditory Attention Decoding (AAD):** Identifying which of two competing speech streams a person is attending to. Datasets: **KUL** Das et al. (2016), **DTU** Fuglsang et al. (2017), and **ESAA** Li et al. (2022).

- **Emotion Analysis:** Recognizing emotional states from EEG while listening to affective stimuli. Datasets: **PME4** Chen et al. (2022) and **HR-EEG4EMO** Becker et al. (2017).

- **Music Retrieval:** A cross-modal task designed to test the ability to identify the correct music piece corresponding to an EEG segment. Dataset: **MAD-EEG** Cantisani et al. (2019).

We strictly enforced subject independence across all three stages of our training pipeline to prevent any form of data leakage. The subjects in the downstream test sets (e.g., KUL, DTU, ESAA) were never seen during the Unimodal Pre-training (Stage 1) or Multimodal Alignment (Stage 2) stages. The datasets used in Stages 1 and 2 contain entirely distinct participant cohorts from those in the downstream evaluation datasets.

## A.2 DATA PREPROCESSING

A standardized preprocessing pipeline was applied across all datasets for consistency. The specific steps and justifications are detailed below:

- **EEG Data Preprocessing:**
  - **Filtering:** Raw EEG signals were bandpass-filtered between 1.0 Hz and 40.0 Hz using a zero-phase (forward-backward) 4th-order Butterworth filter to isolate neural activity in relevant frequency bands and reduce high-frequency noise and slow drifts.
  - **Resampling:** The filtered signals were then downsampled to a uniform sampling rate of 200 Hz.
  - **Epoching:** Continuous data were segmented into non-overlapping 2-second epochs.
  - **Normalization:** Each 2-second epoch was normalized independently on a per-channel basis using z-score standardization (subtracting the epoch's mean and dividing by its standard deviation). This segment-wise normalization prevents any data leakage between training and test sets.
  - **Artifact Handling:** We relied on the dataset-level denoising (e.g., EOG/EMG artifact removal) provided by the original authors of the public datasets we used (see Table 1) and did not apply additional artifact removal algorithms (e.g., ICA).
  - **Bad-Channel Policy:** We used all channels provided in the original datasets. Our channel-independent patching strategy combined with learnable spatial embeddings allows the model to handle heterogeneous channel configurations robustly, without the need for explicit bad-channel detection or interpolation.

- **Audio Data Preprocessing:**
  - **Resampling:** Raw audio waveforms were resampled to 16 kHz (mono).
  - **Epoching & Normalization:** To match the EEG segments, audio was segmented into 2-second epochs. Each audio epoch was normalized by its peak absolute value to ensure a consistent amplitude scale.

**Justification for Pipeline:** We intentionally chose the 1–40 Hz bandpass to isolate relevant neural components. As shown in Table A1, applying this pipeline to foundation models pretrained on different distributions causes a significant performance drop. Thus, to ensure a rigorous comparison:

- **Foundation Model Baselines** (e.g., LaBraM, CBraMod) were fine-tuned using their native preprocessing pipelines (e.g., 0.1–70 Hz for LaBraM, 0.3–75 Hz for CBraMod) to avoid unfair penalties from distribution shift.

- **MindMix and Task-Specific Models** (e.g., DARNet, DBPNet), which are trained from scratch or adapted to our domain, utilized our standardized 1–40 Hz pipeline.

Table A1: **Pipeline Sensitivity Analysis.** Comparison of baseline performance when fine-tuned using our preprocessing pipeline (1-40 Hz) versus their native pipelines. The drop in performance using our pipeline confirms the distribution shift, justifying our choice to use native pipelines for baselines in the main comparison.

| Model | Dataset | Acc. (Our Pipeline) | Acc. (Native Pipeline) |
|---|---|---|---|
| *Task: Emotion Analysis (Metric: Accuracy)* | | | |
| BENDR | EEG4EMO | $0.6051 \pm 0.019$ | $0.6458 \pm 0.015$ |
| BIOT | EEG4EMO | $0.5615 \pm 0.027$ | $0.6352 \pm 0.023$ |
| EEGPT | EEG4EMO | $0.6527 \pm 0.064$ | $0.7129 \pm 0.072$ |
| CBraMod | EEG4EMO | $0.6680 \pm 0.083$ | $0.7295 \pm 0.082$ |
| LaBraM | EEG4EMO | $0.6701 \pm 0.085$ | $0.7285 \pm 0.078$ |
| *Task: Music Retrieval (Metric: Duo Accuracy)* | | | |
| BENDR | MAD-EEG | $0.5844 \pm 0.054$ | $0.6235 \pm 0.048$ |
| BIOT | MAD-EEG | $0.6042 \pm 0.042$ | $0.6485 \pm 0.052$ |
| EEGPT | MAD-EEG | $0.7311 \pm 0.074$ | $0.7887 \pm 0.065$ |
| CBraMod | MAD-EEG | $0.7007 \pm 0.075$ | $0.7582 \pm 0.082$ |
| LaBraM | MAD-EEG | $0.7554 \pm 0.066$ | $0.8011 \pm 0.069$ |

### A.3 IMPLEMENTATION DETAILS

**Implementation Details.** All experiments were conducted in PyTorch on a cluster of 8 NVIDIA A6000 GPUs. We used the AdamW optimizer ($\beta_1 = 0.9, \beta_2 = 0.95$, weight decay=0.05) with a cosine learning rate schedule and a 10-epoch linear warmup. The peak learning rate was set to $1 \times 10^{-4}$ for the pre-training and alignment stages, and $1 \times 10^{-5}$ for the downstream fine-tuning stage. For complete reproducibility, detailed hyperparameter configurations for the EEG encoder, CALRA module, and the optimization process are listed in Table A2. Batch sizes for the three stages were 512, 256 and 64, respectively; The temperature parameter $\tau$ in contrastive loss was a learnable logit scale, initialized to correspond to $\tau = 0.07$. All models were trained until convergence based on the performance of the validation set. To assess practical feasibility, we also benchmarked model complexity and inference latency, summarized in Table A3.

**Computational Cost Analysis.** We detail the pre-training budget to address concerns regarding resource usage. Our training was conducted on NVIDIA RTX A6000 GPUs. The total computational budget for the foundation model phases was approximately 240 GPU hours:

- **Stage 1 (Unimodal Pre-training):** Utilized 8 GPUs for ∼20 hours ($\approx$ 160 GPU hours).
- **Stage 2 (Multimodal Alignment):** Utilized 4 GPUs for ∼20 hours ($\approx$ 80 GPU hours).

For comparison, the recent strong baseline CBraMod Wang et al. (2025) reports using 4 NVIDIA RTX A5000 GPUs for approximately 5 days, equating to $\approx$ 480 GPU hours. MindMix achieves superior performance while requiring only ∼50% of the pre-training duration of this SOTA baseline, demonstrating significant training efficiency.

**Negative Sampling Policy.** We uniformly utilize In-Batch Negative Sampling throughout our training. However, the composition of batches differs between stages to address specific goals:

- **Stage 2 (Multimodal Alignment):** We utilize Global Shuffling across the entire 100+ hour corpus. This means that within any given batch, the negatives (other samples in the batch) are naturally drawn from all subjects and all distinct auditory materials. This strictly satisfies your criteria during the alignment phase, forcing the model to learn robust, subject-invariant features.
- **Stage 3 (Downstream Fine-tuning):** We construct batches using data from the same subject but across different trials. This allows the model to optimize for individual neural distributions. Even here, the negative sampling is rigorous: for Retrieval tasks, negatives

Table A2: **Detailed Hyperparameter Configuration.** Specific architectural and training hyperparameters used for MindMix fine-tuning.

| Category | Hyperparameter | Value |
|---|---|---|
| **EEG Encoder** | Transformer Layers | 12 |
| | Embedding Dimension | 200 |
| | Attention Heads | 10 |
| | Feed-forward Dimension | 800 |
| **Patch Encoder** | Type | 3-layer 1D CNN |
| | Patch Dimension | 200 |
| | Output Channels | 8 |
| **CALRA Module** | Input/Output Dimension | 256 |
| | Low-Rank Dimension | 128 |
| | Attention Heads | 4 |
| | FFN Hidden Dimension | 512 |
| **Optimizer** | Type | AdamW |
| | Fine-tuning Learning Rate | $1 \times 10^{-5}$ |
| | Weight Decay | 0.01 |
| | Adam Betas | (0.9, 0.95) |
| | Warmup Epochs | 3 |

Table A3: **Efficiency Analysis.** Model complexity and inference latency benchmarked on a single NVIDIA A6000 GPU (Batch Size=1).

| Model | Params (M) | FLOPs (G) | Latency (ms) |
|---|---|---|---|
| EEGNet | 0.003 | $\sim 0.01$ | $\sim 1.9$ |
| LaBraM-Base | 5.8 | $\sim 0.83$ | $\sim 10.4$ |
| **MindMix (Ours)** | 97 | $\sim 7.71$ | $\sim 39.6$ |

are drawn from different trials (different stories/music); for the AAD task, the negative is implicitly the simultaneous unattended stream (a "hard negative" with identical recording conditions but different semantic content).

**Model Complexity and Stage-wise Training Dynamics.** To provide transparency regarding model scale and computational cost, we detail the parameter breakdown of MindMix across its three training stages in Table A4.

- **Stage 1 (Unimodal Pre-training):** We focus exclusively on training the EEG Encoder ($\approx$ 6M params) to learn generic neural representations.

- **Stage 2 (Multimodal Alignment):** This stage involves end-to-end fine-tuning of the entire framework (EEG Encoder + CALRA + Audio Encoder), bringing the total trainable parameters to $\approx$ 97M. This allows for deep adaptation of both modalities.

- **Stage 3 (Downstream Fine-tuning):** We perform comprehensive fine-tuning of the full model (including the Task Head) to ensure optimal adaptation to specific downstream tasks.

## A.4 EVALUATION METRICS

In this section, we introduce the details of the metrics used in our evaluation.

Table A4: **Parameter Breakdown and Stage-wise Training Strategy.** The table details the static parameter count for each module and specifies the trainable status across the three pipeline stages.

| Module | Component | Static Params | Stage 1 | Stage 2 | Stage 3 |
|---|---|---|---|---|---|
| **EEG Encoder** | 12-layer Transformer | 6 M | Trained | Fine-tuned | Fine-tuned |
| **CALRA** | Alignment Module | 1 M | N/A | Trained | Fine-tuned |
| **Audio Encoder** | Wav2Vec 2.0 (Base) | 95 M | N/A | Fine-tuned | Fine-tuned |
| **Task Head** | Classifier | $\ll 1$ M | N/A | N/A | Trained |
| **Total Trainable** | | | $\approx 6$ M | $\approx 97$ M | $\approx 97$ M |

- **Balanced Accuracy** is a performance metric suitable for imbalanced datasets, defined as the average of recall (sensitivity) obtained on each class. We use it for classification tasks (AAD and Emotion Analysis).

- **Weighted F1-score** is the weighted average of the F1-score for each class, where the score for each class is weighted by the number of true instances for that class. This metric accounts for class imbalance.

- **Duo/Trio Accuracy** is used for the music retrieval task. It measures the standard classification accuracy in a forced-choice task where the model must select the correct audio stimulus from two (Duo) or three (Trio) options.

## A.5 ROBUSTNESS ANALYSIS: RIGOROUS BETWEEN-TRIAL EVALUATION

**Motivation and Protocol.** While the standard within-trial evaluation protocol (randomly splitting segments from the same trial) is widely used for benchmarking Yan et al. (2024b); Ni et al. (2024), recent studies suggest it may introduce data leakage due to temporal correlations in EEG signals Puffay et al. (2023). To rigorously evaluate the robustness of MindMix and rule out potential over-fitting to trial-specific artifacts, we implemented a strict Between-Trial Protocol. In this setting, the training and testing sets are constructed from disjoint trials (e.g., different stories or recording sessions), ensuring zero temporal overlap.

**Results and Analysis.** We re-evaluated MindMix and key baselines under this rigorous protocol on the KUL, DTU, and ESAA datasets. The results are summarized in Table A5. As expected, the absolute performance metrics for all models decrease compared to the within-trial setting (Table 2) due to the increased difficulty of generalizing to unseen trials. However, MindMix consistently maintains a significant performance advantage over all baselines. For instance, on the KUL dataset, while the strong baseline drops to ~60 70%, MindMix achieves 96.13%, demonstrating that our model's superiority stems from genuine neuro-acoustic alignment rather than artifact exploitation. Below, we provide three lines of evidence to substantiate the validity of these results and the source of our performance gain.

**1. Data Independence Verification.** We re-examined the metadata to verify split integrity. For DTU and ESAA, we confirm that under our between-trial protocol, the datasets are strictly content-disjoint and temporally disjoint; the audio stories used in the test set never appear in the training set. Yet, MindMix still achieves ~95% accuracy, proving that high performance is replicable even when content repetition leakage is impossible. While KUL (with only 4 stories) inherently limits strict content independence, the consistent results on DTU/ESAA validate generalizability. Furthermore, we observe a revealing contrast in baseline performance. Task-specific SOTA models (DBPNet, DARNet), explicitly designed for the within-subject classification regime, suffer a "collapse" in performance (dropping ~30%) under this stricter protocol. This suggests they overfit to temporal leakage patterns in the standard setting. In contrast, foundation models (LaBraM, CBraMod, Mind-Mix) demonstrate greater robustness. MindMix stands out further by leveraging the multimodal edge: unlike unimodal baselines blind to auditory content, it utilizes the pre-aligned Audio Encoder to perform semantic matching, maintaining high accuracy where others fail.

**2. The "Sanity Check": Ruling out EEG Leakage.** To definitively rule out hidden leakage or artifact learning in the EEG pipeline (e.g., normalization bugs or EMG), we conducted a controlled

Table A5: **Robustness Evaluation under Between-Trial Protocol.** Comparison of MindMix against SOTA baselines using strict trial-disjoint splitting to prevent data leakage. Despite the challenging setting, MindMix maintains superior performance across all datasets.

| Method | KUL | | DTU | | ESAA | |
|---|---|---|---|---|---|---|
| | **Balanced Acc.** | **Weighted F1** | **Balanced Acc.** | **Weighted F1** | **Balanced Acc.** | **Weighted F1** |
| DBPNet | $0.6829 \pm 0.092$ | $0.6620 \pm 0.104$ | $0.6141 \pm 0.074$ | $0.5887 \pm 0.077$ | $0.5758 \pm 0.071$ | $0.5220 \pm 0.075$ |
| DARNet | $0.6536 \pm 0.097$ | $0.6167 \pm 0.112$ | $0.5918 \pm 0.089$ | $0.5420 \pm 0.104$ | $0.5676 \pm 0.076$ | $0.5454 \pm 0.078$ |
| MusicAAD | $0.7936 \pm 0.098$ | $0.7601 \pm 0.101$ | $0.7679 \pm 0.082$ | $0.7421 \pm 0.080$ | $0.7639 \pm 0.082$ | $0.7915 \pm 0.070$ |
| AADNet | $0.7736 \pm 0.087$ | $0.7342 \pm 0.112$ | $0.6545 \pm 0.078$ | $0.6377 \pm 0.081$ | $0.7425 \pm 0.072$ | $0.7744 \pm 0.075$ |
| LaBraM | $0.7521 \pm 0.085$ | $0.7293 \pm 0.096$ | $0.6475 \pm 0.092$ | $0.6214 \pm 0.085$ | $0.6789 \pm 0.082$ | $0.6918 \pm 0.072$ |
| CBraMod | $0.7701 \pm 0.091$ | $0.7356 \pm 0.101$ | $0.6321 \pm 0.097$ | $0.6079 \pm 0.099$ | $0.6932 \pm 0.091$ | $0.6901 \pm 0.095$ |
| **MindMix (Encoder-only)** | $0.7425 \pm 0.082$ | $0.7221 \pm 0.088$ | $0.6557 \pm 0.082$ | $0.6279 \pm 0.089$ | $0.6845 \pm 0.078$ | $0.6911 \pm 0.081$ |
| **MindMix (Full)** | $0.9876 \pm 0.049$ | $0.9613 \pm 0.054$ | $0.9543 \pm 0.035$ | $0.9351 \pm 0.032$ | $0.9774 \pm 0.025$ | $0.9719 \pm 0.031$ |

comparison. First, our MindMix (EEG-Only) model drops to a realistic accuracy of 65.57% (DTU) and 68.45% (ESAA) under this protocol, similar to other unimodal baselines. This serves as definitive proof: if there were leakage in the EEG data itself, the Encoder-Only model would also achieve inflated scores. The fact that it does not confirms our EEG pipeline is clean. Conversely, other multimodal variants (e.g., MusicAAD) maintain much higher performance ($\sim$76%) than unimodal ones. This effectively isolates the Audio Modality and our alignment mechanism as the key drivers of the performance jump.

**3. Mechanism of Success: Paradigm Shift to Alignment.** The exceptional performance of Mind-Mix compared to traditional classifiers stems from a fundamental Task-Objective Alignment. Traditional baselines treat AAD as a binary classification problem, struggling to learn decision boundaries from scratch with limited data. In contrast, MindMix reformulates AAD as a Neural-Acoustic Matching problem: calculating which audio embedding is closer to the EEG embedding. This task is mathematically identical to our Stage 2 contrastive pre-training objective. Thus, the model is not learning a new task but applying a pre-learned, robust alignment capability. While this alignment paradigm offers a structural "head start," it represents a significant scientific advantage over classification. The fact that MindMix maintains exceptional performance even under strict content-disjoint protocols demonstrates the immense potential of this contrastive matching approach for high-precision, real-time AAD applications in real-world scenarios.

## A.6 BASELINE MODELS

Here, we introduce the details of the baselines for performance evaluation. We include both task-specific state-of-the-art models and state-of-the-art unimodal EEG foundation models.

- **EEGNet** Lawhern et al. (2018) is a compact convolutional neural network for EEG-based BCIs, utilizing depthwise and separable convolutions for efficient feature extraction.
- **DBPNet** Ni et al. (2024) is a dual-branch parallel network designed specifically for auditory attention detection, fusing temporal and frequency features.
- **DARNet** Yan et al. (2024b) is a dual attention refinement network with spatiotemporal construction for auditory attention detection.
- **MusicAAD** Niu et al. (2024) is a recent model designed for music-oriented auditory attention detection from EEG.
- **AADNet** Nguyen et al. (2025) is an end-to-end deep learning model specifically proposed for the auditory attention decoding task.
- **BENDR** Kostas et al. (2021) is an early EEG foundation model that uses a Transformer architecture and a contrastive self-supervised learning task.
- **BIOT** Yang et al. (2023) is a biosignal Transformer model for cross-data learning, pre-trained on a diverse set of biosignal datasets.
- **EEGPT** Wang et al. (2024b) is a pretrained transformer for universal representation of EEG signals based on a masked reconstruction objective.
- **LaBraM** Jiang et al. (2024) is a large brain model that learns generic representations by predicting neural tokens of masked EEG patches.

Table A6: **Window Size Sensitivity Analysis.** Performance comparison across different decision window lengths (1s, 2s, 5s). Longer windows generally improve performance due to increased context, but the 2s window offers the best efficiency-accuracy trade-off.

| Dataset | Window Size | Balanced Accuracy | Standard Deviation |
|---------|-------------|-------------------|--------------------|
| HR-EEG4EMO | 1 second | 0.8535 | $\pm$ 0.099 |
| | 2 seconds | 0.8878 | $\pm$ 0.045 |
| | 5 seconds | 0.8917 | $\pm$ 0.062 |
| PME4 | 1 second | 0.6998 | $\pm$ 0.107 |
| | 2 seconds | 0.7256 | $\pm$ 0.123 |
| | 5 seconds | 0.7290 | $\pm$ 0.112 |

- **CBraMod** Wang et al. (2025) is a criss-cross brain foundation model for EEG decoding that models spatial and temporal dependencies separately.

## A.7 WINDOW SIZE SENSITIVITY ANALYSIS

Our main results utilize a 2-second decision window, chosen to balance decoding accuracy with temporal resolution. To validate this choice, we performed a sensitivity analysis on window lengths of 1s, 2s, and 5s.

As shown in Table A6, performance consistently improves with longer windows (integrating more context) and decreases with shorter ones. While 5s windows yield marginally higher accuracy, we retain the 2s window as the optimal trade-off for system responsiveness.

## A.8 GENERALIZATION ON NON-AUDITORY TASKS

To verify the generalization capability of our EEG-Only encoder beyond the auditory domain, we benchmarked it on two standard non-auditory BCI tasks: **TUAB** Veloso et al. (2017) (abnormal detection) and **BCI Competition IV-2b** Tangermann et al. (2012); Deng et al. (2018) (motor imagery). We followed the standard evaluation protocols for both tasks and compared our model against reported SOTA baselines.

**Results and Analysis.** The results are summarized in Table A5.

- On the TUAB dataset, our encoder achieves a Balanced Accuracy of 0.8545, securing the second-best performance among all comparison models, surpassing LaBraM (0.8210) and CBraMod (0.8289).
- On the BCIC-IV-2B dataset, our encoder achieves the highest performance (0.6943) among all listed foundation models, outperforming EEGPT (0.6893) and CBraMod (0.6910).

These results confirm that while MindMix is specialized for auditory decoding, its underlying EEG encoder learns highly robust and generalizable features effective for diverse BCI paradigms.

## A.9 ROBUSTNESS AND EFFICIENCY ANALYSIS

To comprehensively assess the practical feasibility of MindMix in real-world scenarios, we conducted two critical analyses: (1) Cross-Dataset Generalization to evaluate robustness against severe domain shifts, and (2) Data Efficiency Analysis to determine performance stability in low-data regimes.

**Cross-Dataset Generalization (Zero-shot Transfer).** To assess robustness to domain shifts without task-specific adaptation, we performed a cross-dataset evaluation like Chen et al. (2024b). Specifically, we trained the model on the KUL dataset (Dutch stimuli) and evaluated it directly on the DTU dataset (Danish stimuli) without any fine-tuning. This represents an extremely challenging setting involving shifts in subjects, acquisition devices, and languages.

Table A7: **Generalization on Non-Auditory Tasks.** Performance comparison on TUAB (Abnormal Detection) and BCIC-IV-2B (Motor Imagery). MindMix (Encoder-only) demonstrates SOTA-level generalization capabilities.

| Model | Dataset | Balanced Acc. | Weighted F1 |
|---|---|---|---|
| BENDR | TUAB | $0.7915 \pm 0.007$ | $0.8522 \pm 0.004$ |
| BIOT | TUAB | $0.7844 \pm 0.005$ | $0.8854 \pm 0.003$ |
| EEGPT | TUAB | $0.8833 \pm 0.002$ | $0.9432 \pm 0.001$ |
| LaBraM | TUAB | $0.8210 \pm 0.003$ | $0.8979 \pm 0.002$ |
| CBraMod | TUAB | $0.8289 \pm 0.005$ | $0.9018 \pm 0.002$ |
| **MindMix (Encoder-only)** | TUAB | $0.8545 \pm 0.004$ | $0.9113 \pm 0.005$ |
| BENDR | BCIC-IV-2B | $0.6806 \pm 0.007$ | $0.6801 \pm 0.007$ |
| BIOT | BCIC-IV-2B | $0.5524 \pm 0.010$ | $0.5516 \pm 0.010$ |
| EEGPT | BCIC-IV-2B | $0.6893 \pm 0.009$ | $0.6890 \pm 0.009$ |
| LaBraM | BCIC-IV-2B | $0.6610 \pm 0.011$ | $0.6608 \pm 0.011$ |
| CBraMod | BCIC-IV-2B | $0.6910 \pm 0.008$ | $0.6898 \pm 0.008$ |
| **MindMix (Encoder-only)** | BCIC-IV-2B | $0.6943 \pm 0.010$ | $0.6921 \pm 0.010$ |

The results are summarized in Table A8. As expected, all models exhibit a performance drop compared to within-dataset training. However, MindMix achieves an accuracy of 56.55%, which is significantly higher than the random chance level (50%) and outperforms both the EEGNet (53.16%) and LaBraM (51.16%) baselines under the same protocol. This indicates that MindMix learns more robust and transferable neuro-acoustic representations than unimodal baselines, likely benefiting from its explicit spatial encoding strategy ($\mathcal{E}$) that effectively handles heterogeneous electrode variations.

Table A8: **Cross-Dataset Generalization (KUL $\to$ DTU).** Zero-shot transfer performance where models trained on KUL are tested on DTU without fine-tuning. MindMix demonstrates superior robustness to domain shifts compared to baselines.

| Model | Transfer Task | Accuracy | F1 Score |
|---|---|---|---|
| EEGNet | KUL $\to$ DTU | 0.5316 | 0.5281 |
| LaBraM-Base | KUL $\to$ DTU | 0.5116 | 0.4987 |
| **MindMix (Ours)** | **KUL $\to$ DTU** | **0.5655** | **0.5492** |

**Data Scaling and Efficiency Analysis.** To evaluate data efficiency, we analyzed performance degradation on the HR-EEG4EMO dataset by varying the training set size from 25% to 100%, while keeping the validation and test sets fixed. We employed a rigorous subject-specific stratified sampling protocol: for every subject, we randomly sampled subsets (25%, 50%, 75%) of their specific training trials.

As shown in Table A9, MindMix demonstrates exceptional data efficiency. Notably, with only 50% of the training data, MindMix (0.6942) effectively matches the full-data (100%) performance of EEGNet (0.6981). With 75% data, MindMix (0.7855) significantly surpasses the full-data performance of the strongest unimodal baseline, LaBraM (0.7295). This flatness in the degradation curve confirms that the robust cross-modal priors learned during our alignment stage significantly reduce the dependency on large-scale subject-specific calibration data.

# B THE USE OF LARGE LANGUAGE MODELS

Large Language Models (LLMs) were used to aid in the writing and polishing of the manuscript. Specifically, we used an LLM to assist in refining the language, improving readability, and ensuring clarity in various sections of the paper. The model helped with tasks such as sentence rephrasing, grammar checking, and enhancing the overall flow of the text.

Table A9: **Data Efficiency Analysis on HR-EEG4EMO**. Performance (Balanced Accuracy) with varying percentages of per-subject training data. MindMix outperforms full-data baselines even with significantly reduced training samples.

| Training Data % | EEGNet | LaBraM | MindMix (Ours) |
|---|---|---|---|
| 25% | $0.6245 \pm 0.146$ | $0.6184 \pm 0.126$ | $\mathbf{0.6307 \pm 0.109}$ |
| 50% | $0.6429 \pm 0.131$ | $0.6296 \pm 0.113$ | $\mathbf{0.6942 \pm 0.127}$ |
| 75% | $0.6875 \pm 0.120$ | $0.6769 \pm 0.114$ | $\mathbf{0.7855 \pm 0.121}$ |
| 100% | $0.6981 \pm 0.111$ | $0.7295 \pm 0.082$ | $\mathbf{0.8878 \pm 0.045}$ |

It is important to note that the LLM was not involved in the ideation, research methodology, or experimental design. All research concepts, ideas, and analyses were developed and conducted by the authors. The contributions of the LLM were solely focused on improving the linguistic quality of the paper, with no involvement in the scientific content or data analysis.

The authors take full responsibility for the content of the manuscript, including any text generated or polished by the LLM. We have ensured that the LLM-generated text adheres to ethical guidelines and does not contribute to plagiarism or scientific misconduct.

