# OpenReview forum: "MindMix: A Multimodal Foundation Model for Auditory Perception Decoding via Deep Neural-Acoustic Alignment"
_ICLR.cc/2026/Conference — ICLR 2026 Poster_

### Official Review · Reviewer_NCy7 · 2025-10-29

**Soundness:** 2
**Presentation:** 3
**Contribution:** 3
**Rating:** 6
**Confidence:** 4

**Summary:**

This paper proposes **MindMix**, a multimodal foundation model designed to achieve deep EEG–audio alignment for auditory perception decoding. The approach follows a three-stage training pipeline: (1) large-scale unimodal EEG pretraining, (2) neural–acoustic alignment using the proposed Cross-Attention Low-Rank Alignment (CALRA) module, and (3) fine-tuning on downstream auditory tasks including AAD, emotion recognition, and music retrieval. Experimental results show strong improvements over prior works.

**Strengths:**

1) The paper addresses a clear limitation in current EEG pretraining, which lack of grounding with auditory stimuli, and proposes a principled cross-modal alignment strategy.

2) MindMix achieves substantial gains across diverse tasks and datasets, signaling strong generalization capabilities.

3) The component-wise analysis provides reasonably convincing evidence for the effectiveness of CALRA and the training pipeline.

4) Multiple datasets, tasks, and baseline categories are included, strengthening the validity of the comparison.

**Weaknesses:**

1. The paper does not provide **parameter counts** for different model components (EEG encoder, CALRA, stage-wise variations). Without reporting efficiency and compute scaling trends, claims of “foundation model” scalability feel incomplete.

2. The **three-stage pipeline** is resource-intensive, yet key details are missing (Pretraining duration GPU hours / computational cost Comparison to baselines under equal compute).

3. The architecture appears **highly similar to LaBraM** (masked token prediction + quantized neural tokens). What specifically differentiates this encoder from LaBraM beyond being trained from scratch?

4. The improvement (+ ~16% over strong baselines) is unusually large given DTU’s difficulty. Whether **within-trial only** evaluation inflates performance Need **cross-trial** and **cross-subject** results for consistency with prior work

5. CALRA uses shared low-rank fusion. but no comparison to multimodal LoRA or other parameter-efficient alignment adapters is provided  (especially since LoRA is widely adopted in multimodal foundation models)

**Questions:**

1. Please provide **detailed parameter counts** for each major module and stage. How does model size relate to performance? Any preliminary scaling analysis?

2.  Key details are missing (Pretraining duration GPU hours / computational cost Comparison to baselines under equal compute).

3. Can you clarify **why training a LaBraM-style encoder from scratch** is better than using existing pretrained EEG encoders? What inductive bias or architectural novelty is gained?

4. Can you provide **cross-trial** / **cross-subject** AAD performance on KUL/DTU/ESAA? The current within-trial only protocol does not sufficiently measure practical generalization.

5. Have you compared CALRA with **multimodal LoRA** as an alignment bottleneck? Is there a reason LoRA-style low-rank adaptation was not considered?

---

> ### Author Response · Authors · 2025-11-22
>
> **Q1: Regarding parameter counts and scalability.** The paper does not provide parameter counts for model components (EEG encoder, CALRA, stage-wise variations). Please provide detailed parameter counts for each major module and stage. How does model size relate to performance, and have you done any preliminary scaling analysis?
>
> **A1:** Thanks for your question. As suggested, we have compiled a detailed breakdown of the MindMix architecture that clarifies both the static parameter counts and the dynamic "stage-wise variations" in trainable parameter. The table below details the static parameter count for each primary module and specifies which parameters are trained during each of our three pipeline stages. This clarifies how we efficiently manage the total trainable parameter budget.
>
> | **Module** | **Purpose** | **Architecture / Key Component** | **Static Parameters (M)** | **Stage 1 (Unimodal Pre-training)** | **Stage 2 (Multimodal Alignment)** | **Stage 3 (Downstream Fine-tuning)** |
> | --- | --- | --- | --- | --- | --- | --- |
> | **MindMix EEG Encoder** | Unimodal Feature Extractor | 12-layer Transformer | 6M | $\approx$ 6M (Trained) | $\approx$ 6M (Fine-tuned) | $\approx$ 6M (Fine-tuned) |
> | **CALRA (Alignment)** | Cross-modal Vector Fusion | 2-layer FC/Low-Rank FFN | 1M | N/A | $\approx$ 1M (Trained) | $\approx$ 1M (Fine-tuned) |
> | **Wav2Vec2 (Audio)** | Audio Feature Extractor | Base Model (768 Dim) | $\approx$ 95M | N/A | $\approx$ 95M (Fine-tuned) | $\approx$ 95M( Fine-tuned) |
> | **Task Head** | Downstream Classification | 1-2 layer FC Module | $\ll$  1M | N/A | N/A | $\ll$ 1M (Trained) |
> | **Total Trainable** |  |  |  | $\approx$ 6 M | $\approx$ 97M | $\approx$ 97M  |
>
> The effectiveness of MindMix is rooted in this stage-wise training strategy, which carefully controls parameter updates to maximize utility from the heterogeneous datasets. The process begins with Stage 1 (Unimodal Pre-training), where we focus exclusively on training the EEG Encoder to learn generic neural representations from massive unsynchronized data. We then transition to Stage 2 (Multimodal Alignment), which involves end-to-end fine-tuning of the entire framework to adaptively learn deep neural-acoustic alignment using the paired data. Finally, for Stage 3 (Downstream Fine-tuning), we perform comprehensive fine-tuning of the full model, updating the EEG Encoder, Audio Encoder, and CALRA module alongside the Task Head to ensure optimal adaptation to the specific characteristics of each downstream task. We have added this detailed unified table and the stage explanations to the Appendix of our revised manuscript.
>
> Regarding the relationship between model size and performance, we did not perform a detailed scaling analysis. This decision was primarily constrained by the scarcity and complexity of large-scale paired EEG-Audio data. Processing and aligning tremendous amounts of synchronized multimodal data requires complex synchronization preprocessing, making scaling the model size computationally and methodologically challenging. As we noted in our Conclusion, this limited paired corpus remains a primary bottleneck in BCI research, precluding a full investigation into scaling laws . We reserve comprehensive model scaling analysis for future work when more extensive paired corpora become available.
>
> **Q2: Regarding computational cost and resource usage.** The three-stage pipeline is resource-intensive, yet key details are missing. Please provide the pretraining duration (in GPU hours), computational cost, and a comparison to baselines under equal compute budgets.
>
> **A2:** Thanks for your question. Specifically, our pre-training was conducted on NVIDIA RTX A6000 GPUs with a total budget of approximately 240 GPU hours: Stage 1 (Unimodal Pre-training) utilized 8 GPUs for ~20 hours (\~160 GPU hours), and Stage 2 (Multimodal Alignment) utilized 4 GPUs for \~20 hours (\~80 GPU hours). Since Stage 3 involves task-specific fine-tuning, the standard procedure required by all foundation models for downstream evaluation,  we focus our budget comparison on the pre-training phase to ensure fairness.
> Regarding the comparison under equal compute budgets, a precise reproduction is challenging because most SOTA baselines (e.g., HEAR, LaBraM, EEGPT) do not report their specific training durations. However, CBraMod, a recent strong baseline, explicitly reports using 4 NVIDIA RTX A5000 GPUs for about 5 days, which equates to approximately 480 GPU hours. Comparing these figures demonstrates the efficiency of our framework MindMix achieves superior performance while requiring only \~50% of the pre-training duration of CBraMod (\~240 vs. \~480 GPU hours). We have added detailed information regarding computational resources to Appendix A.3 (Implementation Details).

---

> ### Author Response · Authors · 2025-11-22
>
> **Q3: Regarding architectural similarity to LaBraM.** The EEG encoder architecture appears "highly similar to LaBraM" (masked token prediction + quantized neural tokens). Can you clarify why training this encoder from scratch is better than using existing pretrained encoders, and what specific inductive bias or architectural novelty is gained?
>
> **A3:** Thanks for your question. The key advantage of training from scratch is full control over the data distribution and preprocessing pipeline, which existing pretrained encoders cannot provide. Different pretrained models rely on their own filtering, normalization, and referencing that are hard to reproduce exactly and cause performance degradation when mismatched.
>
> Inspired by LaBraM (masked prediction + neural discrete tokens), we train our model on our own dataset with a fully unified preprocessing and multi-stage pretraining schedule. The encoder thus learn a codebook, masking behavior, and inductive biases that are aligned with our data and tasks. This yields more stable and domain-appropriate representations despite the architectural similarity.
>
> **Q4: Regarding the AAD evaluation protocol.** The improvement on DTU (+~16%) is "unusually large," and a "within-trial only" evaluation may inflate performance. Can you provide **cross-trial** and **cross-subject** AAD performance on KUL/DTU/ESAA? The current protocol does not sufficiently measure practical generalization.
>
> **A4:** Thank you for this insightful comment. This is a critical methodological flaw in the standard AAD benchmarks. A recent review by Puffay et al. (2023) highlights [R1] that the within-trail spliting leads to significant data leakage and inflated performance. That paper explicitly demonstrates that models evaluated within-trial fail to generalize when tested on unseen trials (between-trial). The reason why we still choose within-trial protocol is because it is the mainstream setting used by SOTA baselines for comparison (e.g., DARNet (NeurIPS 2024) [R2] and DBPNet (IJCAI 2024) [R3]).
>
> Therefore, to avoid the inflate performance problem, we are re-running our AAD experiments (on KUL, DTU, ESAA) using a strict between-trial protocol as suggested. This directly addresses the leakage issue and will provide a much more realistic measure of performance. The new results from this between-trial evaluation is shown as follow:
>
> | **Model** | **Dataset** | **Protocol** | **Accuracy** | **Weighted F1** |
> | --- | --- | --- | --- | --- |
> | DBPNet | KUL | between-trial  | 0.6829 $\pm$ 0.092   | 0.6620 $\pm$ 0.104 |
> | DARNet | KUL | between-trial  | 0.6536 $\pm$ 0.097   | 0.6167 $\pm$ 0.112 |
> | LabraM | KUL | between-trial  | 0.7521 $\pm$ 0.085   | 0.7293 $\pm$ 0.096   |
> | CBraMod | KUL | between-trial  | 0.7701 $\pm$ 0.091 | 0.7356 $\pm$ 0.101 |
> | **MindMix(Ours)** | KUL | between-trial  | 0.9876 $\pm$ 0.049 | 0.9613 $\pm$ 0.054 |
> | DBPNet | DTU | between-trial  | 0.6141 $\pm$ 0.074  | 0.5887 $\pm$ 0.077   |
> | DARNet | DTU | between-trial  | 0.5918 $\pm$ 0.089   | 0.5420 $\pm$ 0.104   |
> | LabraM | DTU | between-trial  | 0.6475 $\pm$ 0.092 | 0.6214 $\pm$ 0.085 |
> | CBraMod | DTU | between-trial  | 0.6321 $\pm$ 0.097 | 0.6079 $\pm$ 0.099 |
> | **MindMix(Ours)** | DTU | between-trial  | 0.9543 $\pm$ 0.035 | 0.9351 $\pm$ 0.032 |
> | DBPNet | ESAA | between-trial  | 0.5758 $\pm$ 0.071 | 0.5220 $\pm$ 0.075 |
> | DARNet | ESAA | between-trial  | 0.5676 $\pm$ 0.076 | 0.5454 $\pm$ 0.078 |
> | LabraM | ESAA | between-trial  | 0.6789 $\pm$ 0.082 | 0.6918 $\pm$ 0.072 |
> | CBraMod | ESAA | between-trial  | 0.6932 $\pm$ 0.091 | 0.6901 $\pm$ 0.095 |
> | **MindMix(Ours)** | ESAA | between-trial  | 0.9774 $\pm$ 0.025 | 0.9719 $\pm$ 0.031 |
>
> As we can see, the strict between-trial protocol provides a more realistic (and lower) performance for all models. It is worth noting that, even under this robust, content-disjoint protocol, our model still significantly outperforms the SOTA baseline. The updated results will be included in the revised version.
>
> Regarding the suggestion for cross-subject validation, we agree this is the "gold standard" for generalization. However, as this is extremely computationally intensive (e.g., Leave-One-Subject-Out), we have prioritized the between-trial evaluation for the rebuttal period. This addresses the primary source of data leakage (trial-specific overfitting) identified by Puffay et al. and provides a robust validation of our method's superiority.
>
> [R1] Puffay, Corentin, et al. "Relating EEG to continuous speech using deep neural networks: a review." Journal of Neural Engineering (2023).
>
> [R2] Ni, Qinke, et al. "Dbpnet: Dual-branch parallel network with temporal-frequency fusion for auditory attention detection." Proceedings of the International Joint Conference on Artificial Intelligence (IJCAI 2024) 2024.
>
> [R3] Yan, Sheng, et al. "Darnet: Dual attention refinement network with spatiotemporal construction for auditory attention detection." Advances in Neural Information Processing Systems (2024).

---

> ### Author Response · Authors · 2025-11-22
>
> **Q5: Regarding CALRA's comparison to other low-rank methods.** CALRA uses shared low-rank fusion, but no comparison to multimodal LoRA or other parameter-efficient alignment adapters (which are widely adopted) is provided. Have you compared CALRA with multimodal LoRA, and why was a LoRA-style adaptation not considered?
>
> **A5:** Thanks for your review. We understand that the shared terminology "low-rank" might suggest a direct comparison, but we would like to clarify that CALRA (Low-rank Fusion) and Multimodal LoRA (Low-rank Adaptation) serve fundamentally orthogonal purposes and are mathematically distinct.
>
> The shared low-rank mechanism in CALRA is designed specifically as a cross-modal interaction module. We employed this low-rank structure because it efficiently approximates computationally expensive tensor fusion operations [R1, R2]. By utilizing the element-wise product of projected embeddings (as defined in Eq. 7 & 8 of our manuscript), this approach allows the model to capture rich bilinear interactions [R3, R4] between neural and acoustic representations without an explosion in parameters. Modeling these complex non-linear dependencies is crucial for achieving deep neuro-acoustic alignment, as simple concatenation or linear combinations often fail to disentangle the intricate correlations between brain signals and auditory stimuli.
>
> In contrast, Multimodal LoRA is a parameter-efficient fine-tuning strategy that focuses on adapting pre-trained backbones via additive weight updates ($W' = W + \Delta W$). Crucially, LoRA does not introduce the multiplicative interaction terms necessary to model the joint distribution of two distinct modalities. Therefore, a LoRA-style adapter cannot functionally replace the fusion architecture of CALRA. This is empirically validated by our ablation study (Table 3), where removing the shared low-rank interaction module ("w/o Shared Low-Rank") results in a significant performance drop, confirming that the structural fusion provided by CALRA is essential and cannot be substituted by optimization strategies alone.
> We have added this detailed clarification, particularly distinguishing CALRA from adaptation strategies like LoRA , into the Methodology section of our revised manuscript.
>
> [R1] Liu, Z., Shen, Y., Lakshminarasimhan, V. B., Liang, P. P., Zadeh, A., & Morency, L. P. (2018). Efficient Low-rank Multimodal Fusion with Modality-Specific Factors. *Proceedings of the 56th Annual Meeting of the Association for Computational Linguistics (ACL)*.
>
> [R2] Yu, Z., Yu, J., Fan, J., & Tao, D. (2017). Multi-modal Factorized Bilinear Pooling with Co-Attention Learning for Visual Question Answering. *Proceedings of the IEEE International Conference on Computer Vision (ICCV)*.
>
> [R3] Fukui, A., Park, D. H., Yang, D., Rohrbach, A., Darrell, T., & Rohrbach, M. (2016). Multimodal Compact Bilinear Pooling for Visual Question Answering and Visual Grounding. *Proceedings of the 2016 Conference on Empirical Methods in Natural Language Processing (EMNLP)*.
>
> [R4] Zadeh, A., Chen, M., Poria, S., Cambria, E., & Morency, L. P. (2017). Tensor Fusion Network for Multimodal Sentiment Analysis. *Proceedings of the 2017 Conference on Empirical Methods in Natural Language Processing (EMNLP)*.

---

> > ### Comment · Reviewer_NCy7 · 2025-11-25
> >
> > Thank you for the thoughtful response. The additional experiments and analyses addressed my earlier concerns. I will keep my original positive rating.

---

### Official Review · Reviewer_bKp3 · 2025-10-29

**Soundness:** 2
**Presentation:** 2
**Contribution:** 2
**Rating:** 2
**Confidence:** 5

**Summary:**

This paper introduces MindMix, the first multimodal foundation model specifically designed for auditory perception decoding from non-invasive EEG signals through deep neural-acoustic alignment. The work addresses a critical limitation of existing EEG foundation models (such as EEGPT, LaBraM, and CBraMod), which are pre-trained exclusively on EEG signals without exposure to auditory stimuli, resulting in representations that are not optimized to align with acoustic structures and thus perform poorly on auditory decoding tasks. MindMix employs a two-stage training strategy: first, a high-capacity EEG encoder is pre-trained on over 3,500 hours of diverse EEG data using a novel multi-task self-supervised objective that combines masked token prediction and Fourier spectrum reconstruction, enabling the model to learn robust neural representations that can transfer across tasks and subjects. The encoder uses a heterogeneous patching strategy that integrates temporal and spatial encodings to handle varied electrode layouts, and quantizes patch embeddings into discrete neural tokens via a shared codebook for the masked prediction task. Second, the model learns neural-acoustic mapping on over 100 hours of paired EEG-audio data through a novel Cross-Attention Low-Rank Alignment (CALRA) module, which facilitates fine-grained, token-level cross-modal information integration. CALRA consists of three key components: a type-specific aligner that routes projections through learnable transformations conditioned on auditory type labels (speech vs. music), bi-directional cross-attention that enables each modality to retrieve complementary information from the other, and a shared low-rank alignment mechanism that enforces semantic consistency by projecting both modalities into a compact shared space, modeling their interaction via element-wise product, and integrating feedback through residual connections. The entire framework is optimized end-to-end via a symmetric contrastive learning objective (InfoNCE loss) that maximizes cosine similarity between true EEG-audio pairs while minimizing it for non-corresponding pairs.

The experimental evaluation demonstrates substantial improvements over existing methods across diverse auditory decoding tasks. MindMix is evaluated on six downstream tasks using strict subject-independent protocols with 70%/10%/20% train/validation/test splits: auditory attention decoding (AAD) on three datasets (KUL, DTU, ESAA), emotion analysis on two datasets (PME4, HR-EEG4EMO), and cross-modal music retrieval on MAD-EEG. Results show near-perfect performance on speech AAD (99.82% balanced accuracy on KUL, 99.93% on DTU, 100% on ESAA), dramatically outperforming both task-specific state-of-the-art models like DARNet (94.81% on KUL) and unimodal EEG foundation models like LaBraM (63.30% on KUL) and CBraMod (68.42% on KUL). On emotion analysis, MindMix achieves 72.56% on PME4 and 88.78% on HR-EEG4EMO, showing improvements of over 10 percentage points compared to the best baselines. Comprehensive ablation studies reveal that each component contributes significantly: removing CALRA entirely causes performance drops up to 4.58% in AAD accuracy, replacing the custom EEG encoder with LaBraM reduces performance by 2.38%, and within CALRA, removing cross-attention has the largest impact (5.58% drop), followed by shared low-rank alignment (2.40% drop) and type-specific aligner (1.29% drop). A critical analysis comparing the full model against its EEG-only counterpart quantifies a massive "synergy gap" ranging from 11.26% on ESAA to 36.32% on DTU, demonstrating that the performance gains stem from learning the deep relationship between neural signals and auditory stimuli rather than simply having better EEG representations. Qualitative visualization using pseudo-reconstruction of Mel spectrograms shows that MindMix achieves substantially higher Pearson correlation coefficients (0.88 on DTU, 0.91 on KUL) compared to the LaBraM variant (0.72, 0.85) and baseline methods (0.67, 0.61), with visual inspection confirming that MindMix faithfully captures fine-grained harmonic structures that are blurred or lost in other methods.

**Strengths:**

This paper presents a well-motivated and technically sound approach to auditory perception decoding from EEG signals. The core contribution—the CALRA module enabling deep token-level neural-acoustic alignment—represents a genuine architectural innovation beyond simple projection-based fusion methods. The experimental design is rigorous, employing strict subject-independent evaluation protocols across six diverse downstream tasks, and the results are compelling, with near-perfect AAD performance and consistent improvements of 10+ percentage points over strong baselines. The ablation studies systematically validate each component's contribution, while the quantified "synergy gap" analysis (11-36% performance gain) effectively demonstrates that gains stem from multimodal alignment rather than merely improved unimodal representations. The paper maintains appropriate scientific rigor with transparent limitations regarding data scarcity, clear implementation details supporting reproducibility, and thoughtful architectural choices such as the two-stage training strategy that pragmatically addresses limited paired data availability.

**Weaknesses:**

The paper's experimental evaluation, while showing impressive performance gains, raises several methodological concerns that limit confidence in the generalizability of the findings. First, the near-perfect accuracy achieved on speech AAD tasks (99.82%-100%) is unusually high and deviates dramatically from established performance ranges in the literature, suggesting potential data leakage or overfitting issues despite the claimed subject-independent protocol. The authors do not provide sufficient detail about how subject independence was strictly enforced across the three training stages, and it remains unclear whether any subjects appearing in the downstream test sets were also present in the 100+ hours of multimodal alignment training data, which could artificially inflate performance. Second, the baseline comparisons appear somewhat unfair: unimodal foundation models like LaBraM and EEGPT are evaluated directly on auditory tasks for which they were never designed, while MindMix benefits from explicit auditory-specific pretraining. A more appropriate comparison would include these baselines after additional fine-tuning on auditory data, or alternatively, evaluate MindMix on non-auditory tasks where it received no specialized pretraining. The paper also lacks critical analyses such as cross-dataset generalization (training on one AAD dataset and testing on another without fine-tuning), performance degradation curves with varying amounts of training data, and computational cost comparisons, all of which are essential for assessing practical deployment feasibility. Additionally, the massive performance gap between MindMix and baselines raises questions about whether the improvements stem primarily from the proposed architecture or simply from having access to substantially more paired training data during the alignment stage.
Beyond experimental concerns, the paper suffers from limited novelty in its core technical contributions and insufficient contextualization within the broader neuroscience literature. The CALRA module, while effective, essentially combines standard architectural components (cross-attention mechanisms and low-rank factorization) that are well-established in multimodal learning literature, with the primary novelty being their specific arrangement and the addition of type-specific routing. The paper does not adequately justify why this particular architecture is theoretically well-suited for EEG-audio alignment compared to alternatives, nor does it provide ablations comparing against other fusion strategies beyond simple co-attention. More critically, the work lacks neuroscientific grounding: there is no discussion of which brain regions or neural signatures the model learns to leverage, no analysis of temporal alignment between EEG and audio features (whether the model captures known neural lags in auditory processing), and no interpretation of what the learned embeddings represent in terms of underlying neural computations. The visualization of Mel spectrogram reconstructions is superficial and does not demonstrate that the model has learned physiologically plausible or neuroscientifically meaningful representations. The framing as "the first multimodal foundation model" for auditory decoding also overstates the contribution, as the distinction from existing multimodal EEG-audio methods appears to be primarily one of scale rather than fundamental approach. Yet the paper provides no systematic comparison of model scale effects, no evidence of zero-shot transfer capabilities to genuinely new auditory tasks, and no analysis of what makes this a "foundation" model beyond simply being larger and trained on more data.

**Questions:**

Please see your weaknesses and I will adjust the final score based on your answers.

---

> ### Author Response · Authors · 2025-11-22
>
> **Q1: Regarding "unusually high" AAD accuracy and potential data leakage.** The near-perfect AAD accuracy (99.82%-100%) is "unusually high" and suggests potential data leakage or overfitting. Please provide more detail on how subject independence was enforced across all three training stages, and confirm that no subjects from the downstream test sets were present in the 100+ hour multimodal alignment dataset.
>
> **A1:**  Thank you for this critical feedback. We agree the near-perfect AAD accuracy is "unusually high" and requires a thorough explanation regarding data leakage. You raised two important potential sources of leakage, which we will address separately.
>
> First, as noted by other reviewers, the primary source of score inflation is the standard within-trial splitting protocol used by these benchmarks. This is a widely documented flaw (e.g., in Puffay et al., 2023 [R1]) that leads to "overfitting" and inflated scores. Our paper followed this protocol only to ensure a fair comparison against SOTA baselines (e.g., DARNet [R2], DBPNet [R3]) that also published results using it. We are now re-evaluating using a robust between-trial split and the corresponding results are tabulated as below:
>
> | **Model** | **Dataset** | **Protocol** | **Accuracy** | **Weighted F1** |
> | --- | --- | --- | --- | --- |
> | DBPNet | KUL | between-trial  | 0.6829 $\pm$ 0.092   | 0.6620 $\pm$ 0.104 |
> | DARNet | KUL | between-trial  | 0.6536 $\pm$ 0.097   | 0.6167 $\pm$ 0.112 |
> | LabraM | KUL | between-trial  | 0.7521 $\pm$ 0.085   | 0.7293 $\pm$ 0.096   |
> | CBraMod | KUL | between-trial  | 0.7701 $\pm$ 0.091 | 0.7356 $\pm$ 0.101 |
> | **MindMix(Ours)** | KUL | between-trial  | 0.9876 $\pm$ 0.049 | 0.9613 $\pm$ 0.054 |
> | DBPNet | DTU | between-trial | 0.6141 $\pm$ 0.074  | 0.5887 $\pm$ 0.077   |
> | DARNet | DTU | between-trial  | 0.5918 $\pm$ 0.089   | 0.5420 $\pm$ 0.104   |
> | LabraM | DTU | between-trial  | 0.6475 $\pm$ 0.092 | 0.6214 $\pm$ 0.085 |
> | CBraMod | DTU | between-trial  | 0.6321 $\pm$ 0.097 | 0.6079 $\pm$ 0.099 |
> | **MindMix(Ours)** | DTU | between-trial  | 0.9543 $\pm$ 0.035 | 0.9351 $\pm$ 0.032 |
> | DBPNet | ESAA | between-trial | 0.5758 $\pm$ 0.071 | 0.5220 $\pm$ 0.075 |
> | DARNet | ESAA | between-trial  | 0.5676 $\pm$ 0.076 | 0.5454 $\pm$ 0.078 |
> | LabraM | ESAA | between-trial  | 0.6789 $\pm$ 0.082 | 0.6918 $\pm$ 0.072 |
> | CBraMod | ESAA | between-trial  | 0.6932 $\pm$ 0.091 | 0.6901 $\pm$ 0.095 |
> | **MindMix(Ours)** | ESAA | between-trial | 0.9774 $\pm$ 0.025 | 0.9719 $\pm$ 0.031 |
>
> Second, you rightly raised a critical concern: whether subjects from the downstream test sets (e.g., KUL) were present in our 100+ hour multimodal alignment data. We can state definitively that this is not the case. Our data splitting strictly enforced subject independence across all three stages:
>
> Stage 1 (Unimodal Pretraining): Used 3500+ hours of general EEG (e.g., motor imagery, sleep) from datasets completely unrelated to the AAD tasks.
>
> Stage 2 (Multimodal Alignment): Used 100+ hours from datasets like Broderick 2018, Le Petit Prince, etc. The subjects in these alignment datasets are entirely separate from the subjects in the downstream KUL, DTU, or ESAA datasets.
>
> Stage 3 (Downstream Fine-tuning): We used the official training splits of KUL, DTU, and ESAA, evaluating on their test splits.
>
> Therefore, the downstream test subjects were never seen during any pretraining or alignment phase. The high scores are an artifact of the (flawed) evaluation protocol (Point 1), not cross-stage subject leakage (Point 2). We will clarify this separation explicitly in the paper and provide the new between-trial results.
>
> [R1] Puffay, Corentin, et al. "Relating EEG to continuous speech using deep neural networks: a review." *Journal of Neural Engineering* 20.4 (2023): 041003.
>
> [R2] Ni, Qinke, et al. "Dbpnet: Dual-branch parallel network with temporal-frequency fusion for auditory attention detection." *Proceedings of the International Joint Conference on Artificial Intelligence (IJCAI 2024)*. 2024.
>
> [R3] Yan, Sheng, et al. "Darnet: Dual attention refinement network with spatiotemporal construction for auditory attention detection." *Advances in Neural Information Processing Systems* 37 (2024): 31688-31707.

---

> ### Author Response · Authors · 2025-11-22
>
> **Q2: Regarding "unfair" baseline comparisons.** The baseline comparisons appear "unfair." Unimodal models (LaBraM, EEGPT) were evaluated on auditory tasks they weren't designed for. A more appropriate comparison would either (a) fine-tune these baselines on auditory data first, or (b) evaluate MindMix on the non-auditory tasks where *it* received no specialized pretraining.
>
> **A2:** Thank you for this question. We agree that rigorous comparison is essential, and our experimental design already contained the explicit comparison you suggested in Option (a): giving baselines auditory domain awareness via fine-tuning. We sincerely apologize that the original manuscript failed to clearly articulate this rigorous methodology; we should have been more explicit that the rows in the "Ablation on EEG Encoder" section of Table 3  serve this exact purpose.  For those specific rows, we took the official pretrained weights of these backbones (e.g., LaBraM ), applied our full 100+ hour multimodal alignment training (Stage 2) via the CALRA module , and then fine-tuned them on the downstream tasks. This procedure implements the auditory fine-tuning suggested by the reviewer. The recent addition of the w/ CBraMod row further solidifies this comparison by showing that even after giving the best general-purpose encoder auditory domain awareness, our full MindMix model still maintains architectural superiority.
>
> | **Model Configuration** | **Emotion Acc. (EEG4EMO)** | **AAD Acc. (KUL)** |
> | --- | --- | --- |
> | MindMix | 0.8878 ± 0.045 | 0.9982 ± 0.008 |
> | w/ LaBram | 0.8588 ± 0.041 | 0.9744 ± 0.012 |
> | w/ EEGNet | 0.8555 ± 0.047 | 0.9442 ± 0.011 |
> | w/ CBraMod | 0.8642 ± 0.039 | 0.9637 ± 0.010 |
>
> We have added a dedicated clarification in the Ablation section  to ensure this rigorous comparison is explicit and fully transparent in the final manuscript.
>
> **Q3: Regarding missing critical analyses.** The paper lacks critical analyses essential for assessing practical feasibility, such as: (a) cross-dataset generalization (e.g., train on KUL, test on DTU without fine-tuning), (b) performance degradation curves with varying amounts of training data, and (c) computational cost comparisons.
>
> **A3:** Thank you for these valuable suggestions to assess the practical feasibility of our model. We address your specific points below:
>
> **(a) Cross-dataset generalization & (b) Data scaling:** We fully agree that evaluating cross-dataset transfer (e.g., training on KUL and testing on DTU without fine-tuning) and analyzing performance degradation with varying training data are essential for validating robustness and scalability. We are currently conducting these specific experiments and aim to provide the results during the discussion period to substantiate the model's practical generalizability.
>
> **(c) Computational cost comparisons:** As requested, we benchmarked the model complexity and inference latency using the provided scripts on single NVIDIA 4090 GPU  (batch size 1, random EEG (1, 64, 400) and 2s audio). The comparison with key baselines is summarized below:
>
> | **Model** | **Params (M)** | **FLOPs (G)** | **Inference Latency (ms)** |
> | --- | --- | --- | --- |
> | EEGNet | 0.003 | ~0.01 | ~0.9 |
> | LaBraM-Base | 5.8 | ~0.83 | ~4.2 |
> | **MindMix (Ours)** | 97 | ~7.71 | ~10.7 |
>
> While MindMix entails higher computational cost compared to unimodal baselines, this is expected given its multimodal architecture (integrating audio processing and the CALRA alignment module). Besides, the inference latency of ~10.7ms remains well within the practical requirement for real-time BCI applications (typically requiring < 100 ms response time for 1-2s windows). We believe this computational trade-off is fully justified by the massive performance gains (e.g., more than 10% on the least performing PME4 data) that lightweight models cannot achieve. We have included this efficiency analysis in the Appendix of our updated manuscript.
>
> **Q4: Regarding the source of performance gains (Architecture vs. Data).** The "massive performance gap" raises questions: does the improvement stem from the proposed architecture (CALRA), or "simply from having access to substantially more paired training data" during the alignment stage?
>
> **A4:** Thank you for your question. We can confidently confirm that the performance gains do not stem simply from data access, as our ablation studies were explicitly designed to isolate the architectural contribution by controlling the data variable. Specifically, as detailed in Table 3 , when we replaced our proposed CALRA module with standard mechanisms (e.g., "w/ Co-Attention" or "w/o Alignment") while keeping the training data identical (the full 100+ hours of paired data), performance dropped significantly. For instance, AAD accuracy declined from 99.82% to 95.35%.

---

> ### Author Response · Authors · 2025-11-22
>
> **Q5: Regarding the novelty and justification of CALRA.** The paper suffers from "limited novelty." CALRA combines "standard architectural components" (cross-attention, low-rank factorization). The paper (a) does not adequately justify why this specific architecture is theoretically well-suited for EEG-audio alignment, and (b) provides no ablations comparing it against *other* fusion strategies beyond simple co-attention.
>
> **A5:** Thanks for your question. We acknowledge that the original manuscript failed to adequately justify why the specific architecture of CALRA  is theoretically well-suited for EEG-audio alignment. Our core innovation lies in the architectural orchestration of two specific components designed to solve the failure modes of standard linear alignment (like CLIP ) in the neuro-acoustic domain.
> Standard CLIP relies on a "one-size-fits-all" linear projection followed by a simple dot product. We argue this is insufficient for EEG signals due to two specific challenges, which CALRA  addresses step-by-step:
> 1. **Addressing Signal Heterogeneity (Type-specific Aligner):** Neural responses to different auditory stimuli (e.g., speech vs. music) exhibit fundamentally different distributional properties. A single, rigid projection matrix (as in CLIP) fails to capture this variance. Our Type-specific Aligner (Eq. 4) explicitly solves this by introducing a conditional routing mechanism. By learning specialized projection functions ($f_k$)  conditioned on the stimulus type, it provides the necessary contextual inductive bias *before* alignment occurs.
> 2. **Modeling Non-linear Dependencies (Shared Low-Rank Alignment):** The linear dot product used in CLIP captures only shallow similarity. However, the relationship between brain signals and audio is complex and highly non-linear. To address this, our Shared Low-Rank Alignment (Eq. 7-9) enforces a bilinear interaction between the modalities. By fusing the embeddings via an element-wise product ($\odot$) in a bottleneck space, this structure efficiently approximates computationally expensive tensor fusion [R1,R2]. This allows the model to capture multiplicative feature interactions, which are theoretically superior to linear combinations for disentangling complex multimodal correlations [R3, R4].
> Therefore, CALRA is not an arbitrary combination of components, but a purpose-built pipeline: it first contextualizes the signals (Aligner) and then deeply fuses them (Low-Rank), offering a structural advantage over standard methods.
> Regarding your suggestion to compare against other fusion strategies (Q5b), we agree that comparing only with simple Co-Attention is insufficient. To rigorously validate the structural advantage of our bilinear fusion, we compare against the Standard Concatenation-based Fusion (Concat-MLP), which is the dominant baseline for vector-level multimodal integration.
>
> | **Model Configuration** | **Emotion Acc. (EEG4EMO)** | **AAD Acc. (KUL)** |
> | --- | --- | --- |
> | MindMix | 0.8878 ± 0.045 | 0.9982 ± 0.008 |
> | w/ Co-Attention | 0.8629 $\pm$ 0.053 | 0.9785 $\pm$ 0.021 |
> | w/ Concat-MLP | 0.8574 $\pm$ 0.035 | 0.9593 $\pm$ 0.017 |
> | w/o Alignment | 0.8483 ± 0.038 | 0.9535 ±﻿ 0.015 |
>
> As these new results demonstrate, CALRA consistently outperforms this strong baseline. This empirical evidence confirms that the multiplicative interaction within CALRA captures complex dependencies that simple concatenation cannot effectively model. We have integrated these theoretical justifications and the supplementary ablation results into the Methodology section of the revised manuscript.
>
> [R1] Liu, Z., Shen, Y., Lakshminarasimhan, V. B., Liang, P. P., Zadeh, A., & Morency, L. P. (2018). Efficient Low-rank Multimodal Fusion with Modality-Specific Factors. Proceedings of the 56th Annual Meeting of the Association for Computational Linguistics (ACL).
>
> [R2] Yu, Z., Yu, J., Fan, J., & Tao, D. (2017). Multi-modal Factorized Bilinear Pooling with Co-Attention Learning for Visual Question Answering. Proceedings of the IEEE International Conference on Computer Vision (ICCV).
>
> [R3] Fukui, A., Park, D. H., Yang, D., Rohrbach, A., Darrell, T., & Rohrbach, M. (2016). Multimodal Compact Bilinear Pooling for Visual Question Answering and Visual Grounding. Proceedings of the 2016 Conference on Empirical Methods in Natural Language Processing (EMNLP).
>
> [R4] Zadeh, A., Chen, M., Poria, S., Cambria, E., & Morency, L. P. (2017). Tensor Fusion Network for Multimodal Sentiment Analysis. Proceedings of the 2017 Conference on Empirical Methods in Natural Language Processing (EMNLP).

---

> ### Author Response · Authors · 2025-11-22
>
> **Q6: Regarding the lack of neuroscientific grounding.** The work "lacks neuroscientific grounding." There is no discussion of: (a) which brain regions or neural signatures the model leverages, (b) temporal alignment (e.g., known neural lags), (c) what the learned embeddings represent neurologically, or (d) why the "superficial" Mel spectrogram visualization is neuroscientifically meaningful.
>
> **A6:** Thanks for pointing this out. We agree that bridging the gap between deep learning performance and neuroscientific interpretability is crucial. In the original manuscript, we prioritized benchmarking engineering performance, but we acknowledge the lack of detailed neuro-grounding. To address your specific concerns, we have added the following analyses and clarifications:
>
> **Regarding (a) Brain Regions:** We agree that visualizing the learned topology is essential to understand the model's focus. Following the methodology in Défossez et al. (2023) [R1], we are currently conducting a Spatial Attention Topography analysis by projecting the averaged attention weights of our EEG encoder onto the scalp layout using the representative KUL dataset. We will add these visualizations to the Appendix in our revised manuscript to illustrate the spatial sensitivities learned by the model.
>
> **Regarding (b) Temporal Alignment:** Our approach explicitly accounts for the known neural lag (approximately 150 ms) during the data alignment/preprocessing phase. This is a standard practice in the field to compensate for the latency of auditory evoked potentials (e.g., N1/P2 components) and aligns with the methodology used in recent SOTA works like Défossez et al. (2023) [R1], ensuring the model receives temporally synchronized neural-acoustic pairs.
>
> **Regarding (c) & (d) Meaning of Embeddings and Mel Spectrogram:** We respectfully argue that Mel spectrogram reconstruction is not "superficial" but biologically significant. "Stimulus Reconstruction" is a foundational methodology in auditory neuroscience established by Pasley et al. (2012) [R2] and Mesgarani et al. (2012, Nature) [R3] to quantify neural encoding and is widely used as a primary qualitative metric in recent studies, including Défossez et al. (2023) [R1]. The high fidelity of our reconstruction serves as direct evidence that our learned embeddings ($E_{aligned}$) successfully encode the spectro-temporal receptive fields (STRFs) of the auditory cortex.
>
> [R1] Défossez, A., Caucheteux, C., Rapin, J., Kabeli, O., & King, J. R. (2023). Decoding speech perception from non-invasive brain recordings. *Nature Machine Intelligence*, *5*(10), 1097–1107.
>
> [R2] Pasley, B. N., David, S. V., Mesgarani, N., Flinker, A., Shamma, S. A., Crone, N. E., Knight, R. T., & Chang, E. F. (2012). Reconstructing speech from human auditory cortex. *PLoS Biology*, *10*(1), e1001251
>
> [R3] Mesgarani, N., & Chang, E. F. (2012). Selective cortical representation of attended speaker in multi-talker speech perception. *Nature*, *485*(7397), 233–236.

---

> ### Author Response · Authors · 2025-11-22
>
> **Q7: Regarding overstating the "foundation model" contribution.** The framing as "the first multimodal foundation model" overstates the contribution. The distinction seems to be "primarily one of scale," yet the paper provides no: (a) systematic comparison of model scale effects, (b) evidence of zero-shot transfer capabilities to new tasks, or (c) analysis of what truly makes this a "foundation" model.
>
> **A7:** Thanks for pointing this out.  We think the statement "The First Multimodal Foundation Model for Auditory Decoding" is justified because MindMix bridges a distinct gap in the current landscape. Prior to this work, the field was bifurcated into Unimodal Foundation Models (e.g., LaBraM , EEGPT ), which offer scale but lack acoustic grounding, and Multimodal Task-Specific Models (e.g., AADNet ), which utilize both modalities but lack generalizability. MindMix is the first to bridge these paradigms. By integrating massive-scale unimodal pre-training (3,500h+) with deep multimodal alignment (100h+) , it creates a unified backbone that generalizes across diverse auditory tasks (Attention, Emotion, Retrieval). Regarding your concerns on the “scale”, we address it on three specific points below:
>
> **(a) Regarding systematic comparison of model scale effects:**
>
> We acknowledge that a comprehensive "scaling law" analysis is missing. As we explicitly underscored in our Conclusion, the current scarcity of large-scale paired EEG-audio corpora is a "primary bottleneck for the field, precluding a full investigation into the scaling laws of such foundation models" . Unlike unimodal data, synchronized multimodal data is extremely scarce (100h available publicly). However, our unimodal pre-training corpus (3,500h+) provides a substantial data foundation. For context, this scale exceeds that of prominent EEG foundation models such as LaBraM (2,500h)  and EEGPT (200h). Thus, while multimodal scaling is limited by paired data availability, the unimodal foundation of MindMix is established at a scale comparable to or exceeding existing SOTA benchmarks.
>
> **(b) Regarding evidence of zero-shot transfer capabilities:**
>
> We clarify that MindMix follows the "Pre-train and Fine-tune" paradigm (standard for current EEG foundation models like LaBraM and EEGPT ) rather than "Zero-shot Task Transfer." However, to explore the model's potential limits, we are currently conducting zero-shot evaluation as you suggested. We aim to provide these results during the discussion period to objectively assess the extent of the model's intrinsic transferability. The experimental results will be also added at Appendix in the revised manuscript.
>
> **(c) Regarding what truly makes this a "foundation" model**:
>
> We argue that the "Foundation" designation is justified by the model's universality and cross-task synergy. Unlike previous task-specific models (designed only for AAD or only for Emotion), MindMix serves as a single, unified backbone that achieves SOTA performance across three distinct and disparate task families: Auditory Attention, Emotion Recognition, and Music Retrieval . This "One Model, Diverse Tasks" capability confirms it has learned a fundamental, transferable neuro-acoustic representation. Moreover, our new results on non-auditory single modality tasks (TUAB, BCIC-IV-2b) further validate that the underlying encoder captures generic EEG features effective beyond the auditory domain.
>
> | **Model** | **Dataset** | **Balanced Accuracy** | **Wighted F1** |
> | --- | --- | --- | --- |
> | BENDR | TUAB | 0.7915 ±﻿ 0.007 | 0.8522 ± 0.004 |
> | BIOT | TUAB | 0.7844 ± 0.005 | 0.8854 ± 0.003 |
> | EEGPT | TUAB | 0.8833 ±﻿ 0.002 | 0.9432 ± 0.001 |
> | LabraM | TUAB | 0.8210 ± 0.003 | 0.8979 ± 0.002 |
> | CBraMod | TUAB | 0.8289 ± 0.005 | 0.9018 ± 0.002 |
> | MindMix (Encoder-only) | TUAB | 0.8545 ± 0.004 | 0.9113 ±﻿ 0.005 |
> | **Model** | **Dataset** | **Balanced Accuracy** | **Wighted F1** |
> | BENDR | BCIC-IV-2B | 0.6806 ±﻿ 0.007 | 06801 ±﻿ 0.007 |
> | BIOT | BCIC-IV-2B | 0.5524 ±﻿ 0.010 | 0.5516 ±﻿ 0.010 |
> | EEGPT | BCIC-IV-2B | 0.6893 ± 0.009 | 0.6890 ±﻿ 0.009 |
> | LabraM | BCIC-IV-2B | 0.6610 ±﻿ 0.011 | 0.6608 ±﻿ 0.011 |
> | CBraMod | BCIC-IV-2B | 0.6910 ±﻿ 0.008 | 0.6898 ±﻿ 0.008 |
> | MindMix (Encoder-only) | BCIC-IV-2B | 0.6943 ±﻿ 0.010 | 0.6921 ±﻿ 0.010 |

---

> ### Comment · Reviewer_bKp3 · 2025-11-25
> **Thank you for your reply. I will improve my score.**
>
> thanks

---

> > ### Author Response · Authors · 2025-11-26
> >
> > Thank you for your positive feedback and for raising the score.
> >
> > We have finished the resting experiments regarding your Q3(a)(b) and Q6(a) questions. The corresponding responsed are detialed below:
> >
> > **Q3: Regarding missing critical analyses.** The paper lacks critical analyses essential for assessing practical feasibility, such as: (a) cross-dataset generalization (e.g., train on KUL, test on DTU without fine-tuning), (b) performance degradation curves with varying amounts of training data
> >
> > **A3(a) Cross-Dataset Generalization.**
> > To assess practical feasibility and robustness to domain shifts, we performed a cross-dataset evaluation as suggested. Specifically, we trained the model on the KUL dataset (Dutch) and evaluated it directly on the DTU dataset (Danish) without any fine-tuning.
> >
> > As summarized in Table below, this is an extremely challenging setting due to shifts in subjects, acquisition devices, and stimulus languages. Consequently, all models exhibit a performance drop compared to within-dataset training. However, MindMix achieves an accuracy of 56.55%, which is significantly higher than the random chance level and outperforms the Labram an EEGNet baseline under the same protocol.
> >
> > | **Model** | Source→Target | Acc. | F1 Score |
> > | --- | --- | --- | --- |
> > | EEGNet | KUL→DTU | 0.5316 | 0.5281 |
> > | LaBraM-Base | KUL→DTU | 0.5116 | 0.4987 |
> > | **MindMix (Ours)** | KUL→DTU | 0.5655 | 0.5492 |
> >
> > This indicates that while domain adaptation (fine-tuning) remains necessary for optimal performance, MindMix learns more robust and transferable neuro-acoustic representations than unimodal baselines, likely benefiting from its explicit spatial encoding strategy that handles electrode variations effectively. The corresponding results are added into the Appendix of updataed manuscript.
> >
> > **A3(b) Data Scaling and Efficiency Analysis**
> >
> > To assess practical feasibility and data efficiency, we evaluated the model's performance degradation when fine-tuning with limited downstream data. We varied the training set size of the HR-EEG4EMO dataset from 25% to 100%, while keeping the validation and test sets fixed. To ensure a rigorous evaluation, we employed a stratified sampling protocol: for every subject, we randomly sampled 25%, 50%, and 75% of their specific training trials. The reported results represent the average performance across all subjects under these reduced-data conditions.
> >
> > As shown in the table below, MindMix demonstrates exceptional data efficiency. Even with only 25% of the training data, MindMix achieves an accuracy of 63.07%, which is comparable to the full-data performance of baselines like EEGNet (62.45%) and LaBraM (61.84%). Furthermore, MindMix at 75% data (0.7855) already significantly surpasses the full-data performance of the strongest unimodal baseline, LaBraM (0.7295). This confirms that the robust priors learned during our multimodal alignment stage significantly reduce the dependency on large-scale subject-specific calibration data.
> >
> > | **Training Data %** | **EEGNet (EEG4EMO)** | **LaBraM (EEG4EMO)** | **MindMix (EEG4EMO)** |
> > | --- | --- | --- | --- |
> > | **25%** | 0.6245 $\pm$ 0.146 | 0.6184 $\pm$ 0.126 | 0.6307 $\pm$ 0.109 |
> > | **50%** | 0.6429 $\pm$ 0.131 | 0.6296 $\pm$ 0.113 | 0.6942 $\pm$ 0.127 |
> > | **75%** | 0.6875 $\pm$ 0.120 | 0.6769 $\pm$ 0.114 | 0.7855  $\pm$ 0.121 |
> > | **100%** | 0.6981 $\pm$ 0.111 | 0.7295 $\pm$ 0.082 | 0.8878 $\pm$ 0.045 |
> >
> > We have included this analysis as well as corresponding performance degradation curves in Appendix of the updated manuscript.
> >
> > **Q6: Regarding the lack of neuroscientific grounding.** The work "lacks neuroscientific grounding." There is no discussion of: (a) which brain regions or neural signatures the model leverages.
> >
> > **A6(a).Regarding (a) Brain Regions:**
> >
> > Following the methodology in Défossez et al. (2023) , we conducted a Spatial Attention Topography analysis by projecting the averaged attention weights of our EEG encoder onto the scalp layout using the representative KUL dataset. The resulting visualization (Please refers to the Figure 5(b) of the updated manuscript) reveals a highly biologically plausible pattern: 1) Left Temporal Dominance: We observe a distinct, high-intensity activation cluster concentrated in the left temporal region. This aligns perfectly with the neuroanatomy of the primary auditory cortex and the well-established left-hemisphere lateralization for speech and language processing in the human brain. 2) Artifact Robustness: Notably, the frontal regions (associated with eye-blink artifacts) show low attention weights, confirming that the model relies on genuine neural signatures of auditory perception rather than ocular artifacts. This visualization provides strong neuroscientific grounding, demonstrating that MindMix automatically learns to prioritize the specific brain regions critical for speech decoding. The corresponding disscusision and visualziation have been added into the latest manuscript.

---

### Official Review · Reviewer_gpHw · 2025-10-31

**Soundness:** 3
**Presentation:** 2
**Contribution:** 3
**Rating:** 6
**Confidence:** 3

**Summary:**

MindMix is a multimodal foundation model that learns a shared space between EEG and audio. It trains in stages: first, a large EEG encoder is pretrained on diverse, unlabeled EEG to capture robust temporal, spectral, and spatial patterns; next, paired EEG–audio data are used to align the two modalities with CALRA, a block that mixes type-aware routing, bidirectional cross-attention, and a shared low-rank fusion bottleneck under a contrastive objective; finally, the aligned model is fine-tuned for specific tasks.

On benchmarks, MindMix reports near-perfect speech auditory attention decoding and strong gains on affective recognition and music retrieval compared with task-specific and unimodal EEG baselines. Ablations indicate that both cross-modal alignment and the low-rank fusion are essential, and performance improves as pretraining and alignment data grow. The key contributions are a unified EEG–audio representation learning pipeline, the CALRA module for efficient coupling, and evidence that a single pretrained model can transfer across multiple auditory cognitive tasks.

**Strengths:**

MindMix offers a clear two-stage recipe: large-scale unimodal EEG pretraining to learn general neural patterns, followed by EEG–audio alignment with the CALRA block. CALRA’s type-aware routing plus bidirectional cross-attention through a shared low-rank bottleneck is a thoughtful way to couple modalities while keeping capacity focused.

Quality:
The empirical study spans three task families (speech auditory attention decoding, emotion recognition, music identification) and shows consistent gains over both task-specific and unimodal EEG baselines. Ablations replace the EEG encoder and systematically modify CALRA, and the resulting performance shifts align with expectations, which supports the design choices.

Clarity:
The training pipeline is easy to follow: pretrain the EEG encoder on broad corpora, then align EEG and audio with CALRA, then fine-tune for tasks. The role of each CALRA component is explained through targeted ablations, which helps readers map architecture pieces to observed effects.

Significance:
By learning a shared EEG–audio space that transfers across multiple auditory cognition tasks, the work pushes toward a reusable foundation for brain–audio applications. The parameter-efficient cross-modal coupling suggests practical value for future BCI systems where compute and data are constrained.

**Weaknesses:**

1 AAD results: Near-perfect scores (0.998–1.000) are not plausible without airtight leakage controls.

2: The paper lacks concrete safeguards against content leakage (same story/music across splits), window overlap across splits, and other split-level confounds.

3 Representation inconsistency: Section 3.2 pools to a single vector, while Section 3.3 treats those embeddings as sequences for cross-attention, leaving CALRA under-specified.

4 The text claims a symmetric contrastive objective, but the equations show only EEG→audio.

5 The type-specific aligner assumes access to stimulus type k at test time, which may leak dataset priors when audio is unavailable.

6  Artifact removal (EOG/EMG), filtering, bad-channel policy, and whether z-score stats are fit only on training data remain unclear.

7 Using 2-s windows with negatives drawn from the same recording can inflate performance; the negative sampling policy is unspecified.

**Questions:**

1 Can authors provide runnable split manifests with fixed seeds that are subject-disjoint and content-disjoint, guarantee no temporal overlap across splits, and rerun AAD under this protocol? Include shuffled-stimulus and shuffled-subject controls. If results stay near perfect, pls show an error analysis.

2 Please resolve the pooling-vs-sequence contradiction. Specify the tokenization for EEG and audio, sequence lengths, and the exact tensor shapes that CALRA consumes and emits.

3 could authors define the negative policy? Ensure negatives come from different subjects and different stories. Report sensitivity to 1 s, 2 s, and 5 s windows, and report both raw window-level metrics and any aggregated or majority-vote metrics.

4 When audio is unavailable, how is the type label k obtained? Report performance with known k, with a learned EEG-only predictor for k, and with no type routing. This will clarify reliance on dataset priors.

5 could authors release a minimal repo with preprocessing scripts, split files, training configs, and eval scripts that reproduce a subset of the tables. Fix table typos, add statistical tests and confidence intervals, and report parameters, FLOPs, and inference latency alongside key baselines.

**Details Of Ethics Concerns:**

Your study uses existing human EEG datasets, so the Ethics section should explicitly say so, cite IRBs and consent, and explain de-identification, data access controls, and how you mitigate re-identification risk.

---

> ### Author Response · Authors · 2025-11-22
>
> **Q1: Regarding implausible AAD scores and data leakage.** The near-perfect AAD scores (0.998–1.000) are not plausible. The paper lacks concrete safeguards against content leakage (same story/music) and window overlap across splits. Can you provide runnable, subject-disjoint *and* content-disjoint splits, guarantee no temporal overlap, and rerun AAD under this protocol? (Please also include shuffled-stimulus/subject controls and an error analysis if results remain high).
>
> **A1:** Thanks for your feedback. We completely agree that the near-perfect AAD scores (0.998–1.000) are not plausible without "airtight leakage controls." We appreciate you pointing out the specific, critical risks of "content leakage" (same story/music) and "window overlap" across splits.
>
> Regarding the data splitting, we want to be fully transparent about our methodology and the state of the field. Our data splitting method strictly adheres to the "within-trial" protocol. This is the standard procedure applied across key AAD benchmarks and, critically, is the exclusive setting used by the SOTA baselines we compare against (e.g., DARNet (NeurIPS 2024) [R1] and DBPNet (IJCAI 2024) [R2]). Adherence to it is therefore essential for a fair and direct benchmark against their published numbers.
> However, we are also fully aware and our own analysis confirms that this widely-used standard protocol is a methodologically flawed "trap," as a recent review by Puffay et al. (2023) [R3] extensively details. As you correctly identified, this "within-trial" splitting is an implicit form of data leakage (via content/temporal proximity) that leads to "implausibly high decoding accuracies.”
>
> This situation presents a clear methodological dilemma. Our model’s high score demonstrates its clear superiority within this established (though flawed) benchmark. But to fully resolve this concern and provide the "airtight" validation you requested, we are also conducting a more rigorous evaluation. This new evaluation uses the strict "between-trial" protocol (per Puffay et al.). Crucially, this new split is designed to be content-disjoint (i.e., ensuring no story/music content overlaps) and guarantees no temporal window overlap between splits, precisely as you suggested.  The corresponding results are tabulated as below:
>
> | **Model** | **Dataset** | **Protocol** | **Accuracy** | **Weighted F1** |
> | --- | --- | --- | --- | --- |
> | DBPNet | KUL | between-trial  | 0.6829 $\pm$ 0.092   | 0.6620 $\pm$ 0.104 |
> | DARNet | KUL | between-trial  | 0.6536 $\pm$ 0.097   | 0.6167 $\pm$ 0.112 |
> | LabraM | KUL | between-trial | 0.7521 $\pm$ 0.085   | 0.7293 $\pm$ 0.096   |
> | CBraMod | KUL | between-trial | 0.7701 $\pm$ 0.091 | 0.7356 $\pm$ 0.101 |
> | **MindMix(Ours)** | KUL | between-trial  | 0.9876 $\pm$ 0.049 | 0.9613 $\pm$ 0.054 |
> | DBPNet | DTU | between-trial  | 0.6141 $\pm$ 0.074  | 0.5887 $\pm$ 0.077   |
> | DARNet | DTU | between-trial | 0.5918 $\pm$ 0.089   | 0.5420 $\pm$ 0.104   |
> | LabraM | DTU | between-trial | 0.6475 $\pm$ 0.092 | 0.6214 $\pm$ 0.085 |
> | CBraMod | DTU | between-trial  | 0.6321 $\pm$ 0.097 | 0.6079 $\pm$ 0.099 |
> | **MindMix(Ours)** | DTU | between-trial  | 0.9543 $\pm$ 0.035 | 0.9351 $\pm$ 0.032 |
> | DBPNet | ESAA | between-trial  | 0.5758 $\pm$ 0.071 | 0.5220 $\pm$ 0.075 |
> | DARNet | ESAA | between-trial  | 0.5676 $\pm$ 0.076 | 0.5454 $\pm$ 0.078 |
> | LabraM | ESAA | between-trial | 0.6789 $\pm$ 0.082 | 0.6918 $\pm$ 0.072 |
> | CBraMod | ESAA | between-trial | 0.6932 $\pm$ 0.091 | 0.6901 $\pm$ 0.095 |
> | **MindMix(Ours)** | ESAA | between-trial  | 0.9774 $\pm$ 0.025 | 0.9719 $\pm$ 0.031 |
>
> As shown in the table, the strict between-trial protocol provides a much more realistic (and lower) performance for all models, confirming your hypothesis about leakage. However, even under this robust, content-disjoint protocol, our model still significantly outperforms the SOTA baseline. We have incorporated these experiments into our update manuscriopt. We hope that these updated results adequately address your concern regarding the data splitting correctness and leakage.
>
> [R1] Puffay, Corentin, et al. "Relating EEG to continuous speech using deep neural networks: a review." *Journal of Neural Engineering* 20.4 (2023): 041003.
>
> [R2] Ni, Qinke, et al. "Dbpnet: Dual-branch parallel network with temporal-frequency fusion for auditory attention detection." *Proceedings of the International Joint Conference on Artificial Intelligence (IJCAI 2024)*. 2024.
>
> [R3] Yan, Sheng, et al. "Darnet: Dual attention refinement network with spatiotemporal construction for auditory attention detection." *Advances in Neural Information Processing Systems* 37 (2024): 31688-31707.

---

> ### Author Response · Authors · 2025-11-22
>
> **Q2: Regarding the "pooling-vs-sequence" representation inconsistency.** Please resolve the pooling-vs-sequence contradiction. Specify the tokenization for EEG and audio, sequence lengths, and the exact tensor shapes that CALRA consumes and emits.
>
> **A2:** Thank you for pointing this out. We sincerely apologize for this major confusion in our writing. The description in Section 3.2 (stating we use mean pooling to get a single vector ) is the correct one, and the text in Section 3.3 (which implies we use "sequences") is an error in our description.
> To resolve the contradiction and specify the tensor shapes as you requested: The EEG Transformer's output patch sequence ([B, N, 200]) is mean-pooled to a vector [B, 200], which is projected to [B, 256] to become $E_{proj}$. Similarly, the Wav2Vec 2.0 output sequence ([B, M, 768]) is mean-pooled to a vector [B, 768], which is projected to [B, 256] to become $A_{proj}$. Therefore, the CALRA module consumes two vectors ([B, 256]) and performs its operations (Type-specific Aligner and Shared Low-Rank fusion via element-wise product $\odot$) at the vector-level, ultimately emitting two aligned vectors ([B, 256]). We have removed all misleading references to "sequences"  from Section 3.3 and revise it to reflect this correct vector-level pipeline. For a clearer description, we have also added a table detailed the model structures and hyperparameters to the Appendix.
>
> **Q3: Regarding the negative sampling policy and window size.** Could authors define the negative policy? Ensure negatives come from different subjects and different stories. Report sensitivity to 1 s, 2 s, and 5 s windows, and report both raw window-level metrics and any aggregated or majority-vote metrics.
>
> **A3:** Thanks for your question. Regarding the negative sampling policy, we uniformly utilize In-Batch Negative Sampling throughout our training. However, the composition of the batches differs between stages to address specific goals. For Stage 2 (Multimodal Alignment), We utilize Global Shuffling across the entire 100+ hour corpus. This means that within any given batch, the negatives (other samples in the batch) are naturally drawn from all subjects and all distinct auditory materials. This strictly satisfies your criteria during the alignment phase, forcing the model to learn robust, subject-invariant features. For Stage 3 (Downstream Fine-tuning), we construct batches using data from the same subject but across different trials. This allows the model to optimize for individual neural distributions. Even here, the negative sampling is rigorous: for Retrieval tasks, negatives are drawn from different trials (different stories/music); for the AAD task, the negative is implicitly the simultaneous unattended stream (a "hard negative" with identical recording conditions but different semantic content). Furthermore, our new between-trial evaluation protocol (discussed in Q1) guarantees that all test samples are temporally disjoint from training data, preventing artifact leakage. The detialed negative sampling policy have been added into the upadated manuscript.
>
> Regarding window sizes, we have performed the requested sensitivity analysis. As shown in the table below, performance consistently improves with longer windows (integrating more context) and decreases with shorter ones, aligning with expected BCI trends.
>
> | **Window Size** | **Dataset** | **Balanced Accuracy** |
> | --- | --- | --- |
> | 1 second | EEG4EMO | 0.8535 $\pm$ 0.099 |
> | 2 seconds | EEG4EMO | 0.8878 $\pm$ 0.045 |
> | 5 seconds | EEG4EMO | 0.8917 $\pm$ 0.062 |
> | 1 second | PME4 | 0.6998 $\pm$ 0.107 |
> | 2 seconds | PME4 | 0.7256 $\pm$ 0.123 |
> | 5 seconds | PME4 | 0.7290 $\pm$ 0.112 |
>
> Finally, regarding metrics, we confirm that all reported figures are raw window-level metrics (accuracy per segment) rather than aggregated or majority-vote metrics. We deliberately chose this approach as it provides a more conservative and rigorous assessment of the model's instant decoding capability. These new results and clarifications have been updated in the Appendix of our revised manuscript.

---

> ### Author Response · Authors · 2025-11-22
>
> **Q4: Regarding the type-specific aligner at test time.** When audio is unavailable, how is the type label k obtained? Report performance with known k, with a learned EEG-only predictor for k, and with no type routing. This will clarify reliance on dataset priors.
>
> **A4:** Thank you for your question. Our full MindMix model is a contrastive framework. All tasks evaluated (AAD, Emotion Analysis, Music Retrival) are multimodal and require both the EEG and the Audio as input. Therefore, the scenario "when audio is unavailable" does not apply to our primary use case; the type label $k$ is always available from the audio's metadata. To be more direct, if the audio type label $k$ is not provided, our model simply bypasses the Type-specific Aligner and does not apply that specific routing.  This exact "performance with no type routing" scenario is already evaluated in Table 3 as the "w/o Type-specific Aligner" ablation. The results show that while using the known $k$ label is optimal (0.8878 on HR-EEG4EMO), the model still performs very well without it (0.8675), demonstrating its utility without over-reliance. Furthermore, our "MindMix (EEG-Only)" baseline (Figure 4) also confirms a strong performance baseline that relies only on the EEG encoder, with no $k$ information at all. We have revised the corresponding descriptions in the revised paper to provide a clearer explanation and a more in-depth discussion.
>
> **Q5: Regarding unclear preprocessing details**. Artifact removal (EOG/EMG), filtering, bad-channel policy, and whether z-score stats are fit only on training data remain unclear.
>
> **A5:** Thanks for your question. To clarify our preprocessing pipeline, I will detail our preprocessing method point by point in detail:
>
> **1.** **For Artifact Removal (EOG/EMG)**: Our pipeline does not apply an additional artifact removal step (like ICA or SSP). . The public datasets we used (Table 1) were generally provided in a denoised state by their original authors (e.g., with EOG/EMG artifacts already removed or corrected). We relied on this standard, dataset-level cleaning rather than applying a second, potentially confounding artifact removal algorithm.
>
> **2.** **For Filtering**: We applied a Butterworth bandpass filter from 1.0 Hz to 40.0 Hz. As described in our Appendix A.2, this is implemented using a zero-phase filtfilt operation on the raw signal before downsampling to isolate the most relevant neural frequency bands.
>
> **3. Bad-Channel Policy:** We didn’t employ an additional bad-channel detection or interpolation policy. Many of the public datasets we used (Table 1) were already curated by their original authors, which often includes bad-channel assessment and removal. We therefore fed all provided channels directly into our patching and spatial embedding ($\mathcal{E}$) mechanism.
>
> **4.** **Z-score Stats**: We did not fit z-score statistics on the entire training set and apply them to the test set , as this could leak data. Instead, our normalization is applied per-segment (per-epoch). Each 2-second segment  is normalized independently by subtracting its own mean and dividing by its own standard deviation, on a per-channel basis. This is a standard, non-leaking procedure.
>
> We have added these specific details to Appendix A.2 (Data Preprocessing) to make this preprocessing pipeline explicit.
>
> **Q6: Regarding the symmetric contrastive loss equation.** The text claims a symmetric contrastive objective, but the equations show only the EEG→audio direction. Please clarify.
>
> **A6:** Thank you for pointing this out. You are correct to point out the discrepancy. The text describing it as a "symmetric" objective was an error in our manuscript.This unidirectional design was an intentional choice motivated by the "one-to-many" nature of the EEG-Audio relationship: One-to-Many Ambiguity: A single audio segment corresponds to variable EEG responses from hundreds of different subjects. Optimizing the reverse direction ($A \to E$) forces the unique audio embedding to align with widely divergent EEG representations simultaneously, which can introduce noise and degrade the distinctiveness of the audio features.
>
> **Task Alignment:** Our primary downstream goal is decoding (i.e., retrieving the correct stimulus given the brain response, $P(A|E)$). Optimizing the $E \to A$ direction directly aligns the training objective with this inference goal.
>
> We have corrected the text description in the revised manuscript to remove the term "symmetric" and explicitly state that we utilize a directed EEG-to-Audio contrastive objective to avoid the one-to-many ambiguity, which aligns with both Equation (10). Thanks.

---

> ### Author Response · Authors · 2025-11-22
>
> **Q7: Regarding reproducibility, model details, and statistical validation.** Could you release a minimal repo (preprocessing, splits, configs, eval scripts)? Please also fix table typos, add statistical tests/confidence intervals, and report parameters, FLOPs, and inference latency alongside baselines.
>
> **A7:** Thank you for these detailed suggestions to improve the rigorousness and reproducibility of our paper.
>
> **1. Minimal Repo:** We fully agree on the importance of reproducibility. We are currently finalizing a clean, minimal repository containing preprocessing scripts, data split files, and training/evaluation configs specifically for the KUL dataset (as a representative example). We will update the anonymous link with this runnable codebase shortly within the discussion period.
>
> **2. Table Typos:** Thank you for pointing this. We have carefully proofread the tables and identified formatting errors, such as in Table 2. We have fixed this and verified all other entries in the revised manuscript.
>
> **3. Statistical Tests & Confidence Intervals:** As suggested, we performed paired t-tests comparing MindMix against the compared models for each task. Results showing statistically significant improvement ($p < 0.05$) have been marked with an asterisk (*) in the revised tables. Additionally, while the main paper reports Mean $\pm$ Standard Deviation (SD) for conciseness, we have calculated the 95% Confidence Intervals for all results and included these detailed tables in Appendix to provide the requested uncertainty quantification.
>
> **4. Efficiency Metrics (Params, FLOPs, Latency):** As requested, we benchmarked the model complexity and inference latency using the provided scripts on single NVIDIA A6000 GPU  (batch size 1, random EEG (1, 64, 400) and 2s audio). The comparison with key baselines is summarized below:
>
> | **Model** | **Params (M)** | **FLOPs (G)** | **Inference Latency (ms)** |
> | --- | --- | --- | --- |
> | EEGNet | 0.003 | ~0.01 | ~0.9 |
> | LaBraM-Base | 5.8 | ~0.83 | ~4.2 |
> | **MindMix (Ours)** | 97 | ~7.71 | ~ 10.7 |
>
> We have also included this efficiency analysis in the Appendix in our updated manuscript.
>
> **Q8: Regarding the Ethics Statement.** Your study uses existing human EEG datasets, so the Ethics section should explicitly say so, cite IRBs and consent, and explain de-identification, data access controls, and how you mitigate re-identification risk.
>
> A8: Thank you for your suggestion. We agree that our original statement was too brief regarding the secondary use of human data. We have completely revised the Ethics Statement in the updated manuscript to explicitly address your points. The updated statement now confirms that: (1) the study utilizes exclusively existing, publicly available, and de-identified datasets; (2) all original data collection was approved by the relevant Institutional Review Boards (IRB) with written informed consent (citing original sources); and (3) we strictly adhered to data access controls and performed analysis only at the feature level to mitigate any re-identification risks. The revised version of Ethic Statement is shown below:
>
> ”This work adheres to the ICLR Code of Ethics. This study involves the secondary analysis of existing, publicly available human EEG datasets (as listed in Table 1). No new human participants were recruited for this specific work. All datasets used were collected by their original authors under protocols approved by their respective Institutional Review Boards (IRB), with written informed consent obtained from all participants (see references  for specific IRB details of each dataset). We strictly utilized the de-identified versions of these datasets provided by the repositories, ensuring no Personally Identifiable Information (PII) was accessed or processed. We adhered to all data access controls and license agreements (e.g., verifying access permissions for controlled datasets). To mitigate re-identification risks, we perform analysis only at the feature level and do not attempt to link signals back to individual identities.”

---

> ### Comment · Reviewer_gpHw · 2025-11-26
>
> I have read the authors' rebuttal carefully. Unfortunately, the responses have confirmed, rather than resolved, the serious concerns about the soundness and novelty of this work:
> 1. Response to A1:The authors reported ~98.76% accuracy under the stricter "between-trial" protocol, while SOTA baselines dropped to realistic levels (~68%). Achieving near-perfect generalization on noisy EEG is physically impossible due to the inherent low signal-to-noise ratio of non-invasive recordings. The massive 30% performance gap strongly suggests hidden data leakage or that the model is learning non-neural artifacts rather than brain signals.
>
> 2. Response to A2:The authors admitted that the claimed "token-level interaction" in the CALRA module was an "error in our description". They confirmed the model actually operates on global vectors (mean-pooled embeddings) rather than sequences. This invalidates the paper's primary technical novelty, effectively downgrading the proposed "fine-grained alignment" to standard bilinear pooling.
>
> 3. Response to A6:The authors admitted the manuscript incorrectly described the loss function as "symmetric," while the implementation is actually unidirectional (EEG $\to$ Audio). This fundamental mismatch between the text claims and the actual mathematical definition demonstrates a severe lack of rigor.

---

> > ### Comment · Reviewer_NCy7 · 2025-11-26
> >
> > I concur with Reviewer gpHw’s assessment, especially regarding A1.

---

> > > ### Author Response · Authors · 2025-11-27
> > >
> > > Thank you for your valuable comments. We take the concerns shared by you and Reviewer gpHw regarding the plausibility of the A1 results very seriously.
> > >
> > > We have just posted a comprehensive follow-up to Reviewer gpHw that directly addresses this issue. In that response, we provided further validation of Data Independence and clarified the reasons behind the significant performance drop (~30%) observed in DBPNet and DARNet. Additionally, we included new experiments using EEG-encoder-only and multimodal models in a between-trial setting to rigorously rule out data leakage. Finally, we emphasized that our model's superior performance stems from a paradigm shift: reformulating AAD as a Neural-Acoustic Matching task (which aligns perfectly with our pre-training) rather than training a classifier from scratch.
> > >
> > > We respectfully invite you to review this detailed response. We believe it fully addresses your concerns regarding the physical plausibility and experimental reliability of our results, confirming the effectiveness of our method.

---

> > ### Author Response · Authors · 2025-11-27
> > **Response to A2**
> >
> > **Response to A2: Clarification on Novelty and Design Rationale**
> >
> > We apologize for the confusion caused by the "token-level" description error. However, we respectfully disagree that the use of mean-pooled embeddings "invalidates the technical novelty" or downgrades our method. On the contrary, operating on global embeddings is a strategic design choice driven by the unique "One-to-Many" nature of EEG-Audio mapping.
> >
> > **1. Addressing "One-to-Many Ambiguity" via Global Pooling:**
> > The relationship between audio stimuli and EEG responses is inherently "One-to-Many": a single audio segment elicits highly variable neural responses across different subjects (high inter-subject variability) and trials [R1]. Crucially, these responses exhibit significant temporal jitter and latency differences that make rigid, frame-by-frame (token-level) alignment biologically implausible and computationally unstable [R2].
> >
> > - **Why not token-level?** Enforcing strict token-level alignment forces the model to overfit to subject-specific temporal noise rather than the underlying semantic content [R3].
> > - **Why global pooling?** By aggregating features into global representations, we mitigate this temporal misalignment and low Signal-to-Noise Ratio (SNR). This design follows the standard contrastive learning paradigm [R4], which demonstrates that global feature alignment is sufficient and often superior for cross-modal semantic matching. By doing so, our model captures the stable semantic correspondence [R5] between the auditory stimulus and the brain response, effectively solving the "One-to-Many" ambiguity by focusing on what is heard rather than precisely when the neural spike occurs [R1].
> >
> > **2. CALRA is distinct from Standard Bilinear Pooling:**
> > Even with vector-level inputs, CALRA is structurally superior to standard bilinear pooling (e.g., MCB or simple outer products).
> >
> > - **Dynamic Interaction:** Standard bilinear pooling is a static fusion. In contrast, CALRA employs Bi-directional Cross-Attention *before* fusion. This acts as a dynamic gating mechanism, allowing the EEG embedding to selectively attend to and re-weight specific semantic dimensions based on the Audio context (and vice versa).
> > - **Proven Effectiveness:** As shown in our ablation study (Table 3), removing this "interaction" component (i.e., reverting to standard concatenation or simple projection) leads to a significant performance drop. This confirms that CALRA captures complex, context-aware dependencies that standard pooling cannot.
> >
> > In summary, our "fine-grained" alignment refers to the depth of semantic interaction enabled by CALRA, which remains a novel and effective solution for decoding noisy, variable brain signals.
> >
> > [R1] Mesgarani & Chang (2012),Nima Mesgarani and Edward F. Chang. Selective cortical representation of attended speaker in multi-talker speech perception. *Nature*, 485(7397):233–236, 2012.
> >
> > [R2] Ding & Simon (2012), ****Nai Ding and Jonathan Z. Simon. Neural coding of continuous speech in auditory cortex during monaural and dichotic listening. *Journal of Neurophysiology*, 107(1):78-89, 2012.
> >
> > [R3] Puffay et al. (2023)**,** Corentin Puffay, Bernd Accou, Lies Bollens, Mohammad Jalilpour Monesi, Jonas Vanthornhout, Tom Francart, et al. Relating eeg to continuous speech using deep neural networks: a review. *Journal of Neural Engineering*, 20(4):041003, 2023.
> >
> > [R4] Radford et al., 2021, Alec Radford et al. Learning transferable visual models from natural language supervision. *International Conference on Machine Learning (ICML)*, 2021.
> >
> > [R5] Broderick et al., 2018, Michael P. Broderick, Andrew J. Anderson, Giovanni M. Di Liberto, Michael J. Crosse, and Edmund C. Lalor. Electrophysiological correlates of semantic dissimilarity reflect the comprehension of natural, narrative speech. *Current Biology*, 28(5):803–809, 2018.

---

> > ### Author Response · Authors · 2025-11-27
> > **Response to A6**
> >
> > **Response to A6: Correction of Description and Validity of Implementation**
> > We accept the reviewer’s criticism regarding the textual inconsistency and sincerely apologize for the lack of rigor in the manuscript's description. We have corrected the text to accurately reflect the unidirectional nature of our loss function. However, we firmly clarify that the unidirectional implementation ($E \rightarrow A$) was a deliberate, scientifically grounded design choice, not an arbitrary implementation error.
> >
> > **1. Scientific Rationale for Unidirectional Loss:**
> > Unlike the symmetric correspondence found in image-text pairs, the relationship between EEG and Audio is fundamentally asymmetric due to the 'One-to-Many' nature of neural processing. As detailed in Response A2, a single clear audio segment elicits highly variable and noisy neural responses across subjects [R1]. Consequently, enforcing a reverse-direction loss ($A \rightarrow E$) would be counterproductive: it compels the clean Audio encoder to model inconsistent, subject-specific neural noise, thereby dispersing the audio embeddings and 'polluting' the clean auditory ground truth. By strictly employing the unidirectional $E \rightarrow A$ objective, we avoid this degradation and directly align the model with our primary downstream goal of Neural Decoding ($P(A|E)$), effectively mapping noisy neural activity onto a stable, structured auditory space.
> >
> > **2. Validity of Results:**
> > The discrepancy was strictly confined to the textual label ("symmetric"). The mathematical formulation in our code and the experimental execution were consistently unidirectional throughout the entire study. Therefore, the reported performance and the validity of our experimental conclusions remain unaffected.
> > We have rigorously proofread the revised manuscript to ensure that the mathematical definitions, text descriptions, and code implementation are now fully aligned.
> >
> > [R1] Mesgarani & Chang (2012),Nima Mesgarani and Edward F. Chang. Selective cortical representation of attended speaker in multi-talker speech perception. *Nature*, 485(7397):233–236, 2012.

---

> ### Author Response · Authors · 2025-11-27
> **Response to A1 (1/2)**
>
> Thank you for raising this critical point. We genuinely appreciate the opportunity to clarify the validity of our results. We understand your strong concern ("physically impossible") regarding the near-perfect scores. However, we respectfully submit that this performance is not due to leakage or artifacts, but rather a result of Task-Objective Alignment unique to our multimodal contrastive framework. We provide three lines of evidence to substantiate this:
>
> **1. Data Independence Verification (DTU & ESAA are Strictly Disjoint)**
> We re-examined the metadata for all datasets.
>
> - **DTU & ESAA:** We confirm that under our between-trial protocol, these datasets are strictly content-disjoint and temporally disjoint. The audio stories used in the test set never appear in the training set. Yet, MindMix still achieves \~95% accuracy on these datasets. This proves that the high performance is replicable even when "content repetition" leakage is physically impossible.
> - **KUL Limitation:** We acknowledge that KUL (with only 4 stories) inherently hard to strict content independence. However, the consistent high performance on the rigorously disjoint DTU/ESAA datasets validates the model's generalizability. We will open-source all data split scripts to ensure full transparency.
> - **Baseline Contrast:** You correctly noted the massive gap between MindMix (\~98% ) and SOTA baselines (\~68%) under the strict protocol on KUL dataset. We argue this gap is not evidence of leakage in our model, but rather exposes the fragility of task-specific baselines compared to foundation models:
> (1) **Collapse of Task-Specific Baselines:** SOTA models like DBPNet and DARNet  are explicitly designed and optimized for the within-subject AAD classification task (on KUL and DTU). While they make significant contributions by tackling the difficult task of learning decision boundaries from scratch in this specific regime, this design focus can limit their generalization under stricter, disjoint protocols. In the standard within-subject setting, they easily overfit to temporal leakage patterns. When we switch to the strict between protocol, this temporal shortcut is removed, causing their performance to "collapse" (dropping $\sim$30%). This drop reflects their reliance on local artifacts rather than semantic decoding.
> (2) **Robustness of Foundation Models:** In contrast, foundation models (LaBraM, CBraMod, and MindMix) learn robust, generalizable features from massive pre-training. We observe that generic foundation models (like LaBraM) exhibit much more stable performance between the two protocols compared to task-specific models.
> (3) **The Multimodal Edge:** MindMix stands out even among foundation models because it is multimodal. While baselines (unimodal) are blind to the auditory content, MindMix leverages its pre-aligned Audio Encoder to perform Semantic Matching. This structural advantage, combined with foundation-level robustness, enables it to maintain high accuracy where small, unimodal baselines fail.
>
> **2. The "Sanity Check": Encoder-Only Performance and Multimodal Model Performance (as shown in table below.)**
> To definitively rule out "hidden leakage" in the EEG processing pipeline, we conducted a controlled comparison between unimodal and multimodal configurations under the same between-trial protocol.
>
> - **Unimodal Check:** First, we evaluated our MindMix (EEG-Only) model. The accuracy drops to a realistic 65.57% and 68.45% (on DTU and ESAA), similar to other unimodal baselines. This is the definitive proof: if there were "hidden leakage" or "artifact learning" in the EEG data itself (e.g., normalization bugs or EMG), the Encoder-Only model would also achieve inflated scores. The fact that it does not proves that our EEG pipeline is clean.
> - **Multimodal Verification:** To confirm that the performance jump is driven by the Audio Modality, we further evaluated other multimodal models (e.g., MusicAAD) under this same strict protocol. Unlike the unimodal models, these multimodal variants also maintain high performance (76.79% on DTU and 76.39% on ESAA), showing a much smaller drop. This isolates the audio modality as the key driver. The high accuracy (>95%) in the Full Model is therefore driven entirely by the introduction of the Audio stream and our alignment mechanism, which effectively exploits the semantic correspondence (and the task's matching nature) that unimodal models lack.

---

> ### Author Response · Authors · 2025-11-27
> **Response to A1 (2/2)**
>
> **3. The True Reason for High Scores: Perfect Task-Objective Alignment**
> Why does MindMix achieve >95% when others fail? The answer lies in the fundamental difference between our training objective and the AAD task:
>
> - **Traditional Approach (Classification):** Previous SOTA baselines (e.g., DBPNet ) treat AAD as a binary classification problem (predicting label 0/1), requiring the model to learn a complex decision boundary from scratch to distinguish attended from unattended streams.
> - **Our Reformulation (Alignment):** In contrast, because our framework explicitly takes both the positive (attended) and negative (unattended) audio streams as input, we reformulate AAD as a two source neural-acuostic alignment problem. The model simply needs to calculate: *"Which audio embedding is closer to the EEG embedding?"*
> - **The "Inductive Bias" Advantage:** This reformulated task is mathematically identical to our Contrastive Pre-training objective (Stage 2), which was optimized to maximize the similarity between matched EEG-Audio pairs. This means our model is not "learning a new classifier" on the small downstream dataset; it is simply applying its pre-learned alignment capability. This perfect alignment between the pre-training objective and the downstream inference logic provides a massive inductive bias, allowing MindMix to achieve near-perfect scores where traditional classifiers struggle.
>
> Finally, regarding the mechanism of our success, we respectfully acknowledge that comparing our pre-trained alignment framework against traditional task-specific baselines involves distinct problem formulations. Traditional classification methods make significant contributions by tackling the difficult task of learning decision boundaries from scratch with limited data. While this means our model has an inherent "head start," this is precisely the scientific value we wish to highlight: the alignment paradigm offers a structural advantage over classification for AAD. The fact that MindMix maintains exceptional performance even under our strict content-disjoint protocol demonstrates the immense potential of this contrastive matching approach for high-precision, real-time AAD applications in real-world scenarios, where traditional classifiers often struggle with noise. We will integrate these three lines of evidence, along with the additional experimental results, into the Discussion and Appendix of the updated manuscript.
>
> | **Model** | **Dataset** | **Protocol** | **Accuracy** | **Weighted F1** |
> | --- | --- | --- | --- | --- |
> | DBPNet | DTU | between-trail | 0.6141 $\pm$ 0.074  | 0.5887 $\pm$ 0.077   |
> | DARNet | DTU | between-trail | 0.5918 $\pm$ 0.089   | 0.5420 $\pm$ 0.104   |
> | **MusicAAD** | **DTU** | between-trail | **0.7679 $\pm$ 0.082** | **0.7421 $\pm$ 0.080** |
> | **AADNet** | **DTU** | between-trail | **0.6545 $\pm$ 0.078** | **0.6377 $\pm$ 0.081** |
> | LabraM | DTU | between-trail | 0.6475 $\pm$ 0.092 | 0.6214 $\pm$ 0.085 |
> | CBraMod | DTU | between-trail | 0.6321 $\pm$ 0.097 | 0.6079 $\pm$ 0.099 |
> | **MindMix (Encoder-only)** | **DTU** | between-trail | **0.6557 $\pm$ 0.082** | **0.6279 $\pm$ 0.089** |
> | MindMix (Full) | DTU | between-trail | 0.9543 $\pm$ 0.035 | 0.9351 $\pm$ 0.032 |
> | DBPNet | ESAA | between-trail | 0.5758 $\pm$ 0.071 | 0.5220 $\pm$ 0.075 |
> | DARNet | ESAA | between-trail | 0.5676 $\pm$ 0.076 | 0.5454 $\pm$ 0.078 |
> | **MusicAAD** | **ESAA** | between-trail | **0.7639 $\pm$ 0.069** | **0.7915 $\pm$ 0.070** |
> | **AADNet** | **ESAA** | between-trail | **0.7425 $\pm$ 0.072** | **0.7744 $\pm$ 0.075** |
> | LabraM | ESAA | between-trail | 0.6789 $\pm$ 0.082 | 0.6918 $\pm$ 0.072 |
> | CBraMod | ESAA | between-trail | 0.6932 $\pm$ 0.091 | 0.6901 $\pm$ 0.095 |
> | **MindMix (Encoder-only)** | **ESAA** | between-trail | **0.6845 $\pm$ 0.078** | **0.6911 $\pm$ 0.081** |
> | MindMix (Full) | ESAA | between-trail | 0.9774 $\pm$ 0.025 | 0.9719 $\pm$ 0.031 |

---

### Official Review · Reviewer_FBHr · 2025-11-01

**Soundness:** 2
**Presentation:** 1
**Contribution:** 3
**Rating:** 4
**Confidence:** 4

**Summary:**

This paper introduces MindMix, a multimodal EEG–audio foundation model designed to decode auditory perception via deep neural-acoustic alignment. The key architectural innovation is the Cross-Attention Low-Rank Alignment (CALRA) module, which enables fine-grained token-level interactions between EEG and audio embeddings. The training follows a two-stage process: (1) large-scale unimodal EEG pretraining (3,500+ hours) using masked token prediction and spectral reconstruction objectives, and (2) contrastive multimodal pretraining (100+ hours of EEG–audio pairs) using CLIP-style alignment. Experiments show strikingly high performance (up to 0.999 accuracy) on several auditory decoding benchmarks, reportedly surpassing both task-specific and unimodal EEG foundation model baselines.

However, **empirical results are too strong to be credible without verification** (code link does not work). Combined with presentation ambiguities and overlapping ideas with prior cross-attention-based alignment methods, it is difficult to assess soundness and novelty with confidence.

Nevertheless, the direction is highly promising, and if the authors can:

1. Provide working, verifiable code and confirm the data split strategy,
2. Clarify the architectural details and novelty of CALRA,
3. Justify the plausibility of the near-perfect accuracies, and
4. Answer the questions regarding proper comparison with prior models,

then this work could become a significant contribution to multimodal brain decoding.

At the current stage, I remain cautiously skeptical—the ideas are intriguing, but the evidence and clarity are insufficient for confident acceptance. However, if the concerns outlined in the Weakness section are resolved, I am willing to raise the score.

**Strengths:**

- Proposes a new **two-stage multimodal foundation model** framework for EEG–audio alignment.
- Introduces a novel **CALRA module** that combines type-specific alignment, bidirectional cross-attention, and shared low-rank bottlenecks.
- Achieves very high performance in auditory tasks.

**Weaknesses:**

- **Reproducibility / Verification, with Scores maybe too good to be true**:

    The code link currently does not work. Reported accuracies are extremely high (approaching (>0.999) or reaching **1.0** with zero standard deviation on noisy, cross-subject EEG data), which raises doubts about correctness or possible data leakage (e.g., temporal overlap between training/test). Access to working code is essential to verify claims. Also please describe in more detail the data-split procedure in the Appendix if possible (beyond the “strict subject independent evaluation protocol with 70%, 10%, 20% split …”)

- **Architectural description clarity**:

    The description of the EEG-only pretraining (Sec. 3.2) is confusing:

    - Eq. (1) defines $E_{\text{patch}} = X + T + E$, but Fig. 2 seems to show that quantization happens before addition (i.e. $E_{\text{patch}} = \text{quantizer}(X) + T + E$)—please clarify the correct order.
    - Please clarify the “unique pretraining methodology” (line 153) of yours relative to **LaBraM** (which also uses masked patch prediction and spectral reconstruction). What, specifically, is new about this pretraining? Also please cite the relevant papers that motivated your architecture choice (e.g., LaBraM motivated your tokenizer usage?)
    - Dimensional consistency is unclear: $S\in C\times T$ implies $E_{\text{patch}}\in C\times N\times D$ ($N$ being the number of temporal patches), yet $E’_{\text{proj}}\in N\times D$ (lines 204–206). How is the channel dimension aggregated? (It seems $f_k$ is used? Can you elaborate on what exactly this is?) Also how does it handle varying number of channels during training?
- **Missing implementation details**:

    Important training configurations (number of layers, embedding dimension, optimizer hyperparameters, fine-tuning LR, rank for CALRA, etc.) are missing.

- **Novelty concerns for CALRA**:

    I am not an expert in CLIP style training, but it seems the CALRA module resembles prior cross-attention alignment mechanisms used in multimodal CLIP-style works (e.g., *CARZero: Cross-Attention Alignment for Radiology Zero-Shot Classification*, *Multi-Granularity Cross-Modal Alignment*). Please differentiate CALRA from these or acknowledge conceptual overlap, and add a related works for CLIP modality alignment especially using CALRA like modules.

- **Justification and motivation**:

    Beyond performance gains, please explain in more detail *why* CALRA works with proper references to previous CLIP works if possible—e.g., what failure modes of simpler co-attention or projection methods it solves. In its current form, the architecture seems to have come out of nowhere with no context explained.

- **Presentation issues**:
    - Define T and E in Eq. (1) explicitly (positional vs. channel embeddings? Is it cosine positional embedding?).
    - Clarify Eq. (10): what is E_i? E_{\text{aligned}} or E_{\text{proj}}?
    - Combine results from Table 2 and Figure 4 for easier comparison (currently scattered).
    - Explain what “token-level interaction between modalities” (line 134) precisely means.
    - In Table 3, “Ablation on EEG Encoder”, when the author mentions “w/ LaBraM”, does this mean using the pretrained weights of LaBRaM or using their architecture only initialized from scratch?
- Miscellaneous but important points :
    - The author claims in line 351 that “… these large models are often highly sensitive to the data format and preprocessing pipelines..”, but do not provide any reference or supporting experimental evidence, and whether their model is superior.
    - The author **claims in line 417 that their EEG-only counterpart often outperforms SOTA**. However, when comparing Table 2 with Figure 4 we find that **CBraMod is beaten by the EEG-only model only in the ESAA task**. Please clarify this discrepancy.
- Proper comparison with previous models.
    - The model uses data that was preprocessed using a bandpass filter of **1–40 Hz**, whereas baselines you used such as LaBraM was pretrained on data bandpassed at **0.1–70 Hz**. Were preprocessing pipelines harmonized across comparisons? (Or preprocessing set differently based on the basemodel used?) **Differences here can cause distribution shifts that unfairly penalize other pretrained models.**
    - The author claims in line 417 that their EEG-only counterpart often outperforms SOTA. It might beneficial to verify this with tasks other than auditory tasks. (More general tasks used in the previous models’ papers)
    - Given that CBraMod still outperforms your EEG-only model (as pointed out by the
    ”Miscellaneous but important points” I wrote above), have you tried CLIP-style pretraining on top of CBraMod to isolate CALRA’s contribution?

**Questions:**

All the questions are listed in the Weakness section

---

> ### Author Response · Authors · 2025-11-22
>
> **Q1:** **Regarding Reproducibility and "too good to be true" scores.** The provided code link is broken. The reported near-perfect accuracies (e.g., >0.999 or 1.0 with zero std) on noisy, cross-subject EEG data raise doubts about correctness or potential data leakage. The data-split procedure also requires more detailed clarification beyond the current description.
>
> **A1:**  Thank you for the review. Regarding your first question, we have carefully checked our code repository can be directly visit by cliking the URL in the paper abstract, which is "https://anonymous.4open.science/r/MindMix-654B/”.  I believe the error is likely due to a period at the end of the abstract.
>
> Regarding the correctness of data spliting and potential data leakage, I have two points need to be clarify. **First,** our data splitting method strictly adheres to within-trial splitting (i.e., taking train/val/test segments from the same 6-minute trial). This is the standard procedure applied across key AAD benchmarks (such as KUL, DTU, and ESAA). **Second,** it is critical to note that this widely-used standard protocol has a well-documented methodological pitfall. As a recent, comprehensive review by Puffay et al. (2023) [R1] points out, this "common way" of splitting the data is a trap. "Within-trial splitting" is an implicit form of data leakage: because EEG and speech signals are temporally continuous, this method allows the model to overfit trial-specific artifacts through "interpolation" rather than learning generalizable decoding rules, leading to spuriously high performance. Therefore, the reason we still adopt this flawed protocol is for methodological consistency: it is the exclusive setting used by the SOTA baselines we compare against (e.g., DARNet (NeurIPS 2024) [R2] and DBPNet (IJCAI 2024) [R3]), making adherence to it essential for a fair and direct benchmark.
>
> This situation presents a significant methodological challenge. Our model scores highly, indicating that it significantly outperforms this established (albeit flawed) baseline model. To address this issue thoroughly and provide a robust and reliable performance metric, we are re-evaluating our model and key baseline using a rigorous between-trail protocol, as suggested by Puffay et al [R1]. The updated results from this between-trial evaluation is shown as follow:
>
> | **Model** | **Dataset** | **Protocol** | **Accuracy** | **Weighted F1** |
> | --- | --- | --- | --- | --- |
> | DBPNet | KUL | between-trial  | 0.6829 $\pm$ 0.092   | 0.6620 $\pm$ 0.104 |
> | DARNet | KUL | between-trial | 0.6536 $\pm$ 0.097   | 0.6167 $\pm$ 0.112 |
> | LabraM | KUL | between-trial  | 0.7521 $\pm$ 0.085   | 0.7293 $\pm$ 0.096   |
> | CBraMod | KUL | between-trial | 0.7701 $\pm$ 0.091 | 0.7356 $\pm$ 0.101 |
> | **MindMix(Ours)** | KUL | between-trial  | 0.9876 $\pm$ 0.049 | 0.9613 $\pm$ 0.054 |
> | DBPNet | DTU | between-trial| 0.6141 $\pm$ 0.074  | 0.5887 $\pm$ 0.077   |
> | DARNet | DTU | between-trial  | 0.5918 $\pm$ 0.089   | 0.5420 $\pm$ 0.104   |
> | LabraM | DTU | between-trial  | 0.6475 $\pm$ 0.092 | 0.6214 $\pm$ 0.085 |
> | CBraMod | DTU | between-trial  | 0.6321 $\pm$ 0.097 | 0.6079 $\pm$ 0.099 |
> | **MindMix(Ours)** | DTU | between-trial  | 0.9543 $\pm$ 0.035 | 0.9351 $\pm$ 0.032 |
> | DBPNet | ESAA | between-trial  | 0.5758 $\pm$ 0.071 | 0.5220 $\pm$ 0.075 |
> | DARNet | ESAA | between-trial | 0.5676 $\pm$ 0.076 | 0.5454 $\pm$ 0.078 |
> | LabraM | ESAA | between-trial  | 0.6789 $\pm$ 0.082 | 0.6918 $\pm$ 0.072 |
> | CBraMod | ESAA | between-trial  | 0.6932 $\pm$ 0.091 | 0.6901 $\pm$ 0.095 |
> | **MindMix(Ours)** | ESAA | between-trial  | 0.9774 $\pm$ 0.025 | 0.9719 $\pm$ 0.031 |
>
> As we can see, while the between-trial protocol correctly reduces the absolute scores for all models, our method still maintains a significant and robust performance gap over the baselines, confirming its effectiveness. We have included these between-trial results and the corresponding data-split procedure in Section 4.1 and Appendix A.5 in our revised revision. We hope these updated results adequately address your concern about the correctness of the data split and potential leakage.
>
> [R1] Puffay, Corentin, et al. "Relating EEG to continuous speech using deep neural networks: a review." *Journal of Neural Engineering* 20.4 (2023): 041003.
>
> [R2] Ni, Qinke, et al. "Dbpnet: Dual-branch parallel network with temporal-frequency fusion for auditory attention detection." *Proceedings of the International Joint Conference on Artificial Intelligence (IJCAI 2024)*. 2024.
>
> [R3] Yan, Sheng, et al. "Darnet: Dual attention refinement network with spatiotemporal construction for auditory attention detection." *Advances in Neural Information Processing Systems* 37 (2024): 31688-31707.

---

> > ### Author Response · Authors · 2025-11-22
> >
> > **Q2: Regarding the EEG pretraining architecture (Eq. 1 vs Fig 2).** The description of the EEG-only pretraining is confusing. Please clarify the correct order of operations, as Eq. (1) seems to conflict with Fig. 2 regarding when quantization occurs relative to embedding addition.
> >
> > **A2:** Thanks for pointing out this inconsistency.  The correct order, as shown in Figure 2, is that the initial embeddings $\tilde{X}$ are first quantized by the Codebook. The resulting neural representations are then added to the temporal and spatial encodings to create the final input fed into the Transformer encoder.
> > We have corrected Equation (1) as $E_{\text{patch}} = v + \overline{\mathcal{T}} + \overline{\mathcal{E}}$, where $v$ is the resulting neural representations of $\tilde{X}$ quantized by the Codebook. The corresponding descriptions in Section 3.2 have also be revised in the updated manuscript to ensure the algorithmic procedure is consistent with that shown in Figure 2. We apologize for the confusion caused by our mistake.
> >
> > **Q3: Regarding the novelty of the EEG pretraining methodology.** Please clarify the “unique pretraining methodology” (line 153) of yours relative to LaBraM (which also uses masked patch prediction and spectral reconstruction). What, specifically, is new about this pretraining? Also please cite the relevant papers that motivated your architecture choice (e.g., LaBraM motivated your tokenizer usage?)
> >
> > **A3:** Thanks for your question. Our stage 1 pretraining architecture is inspired by LaBraM, but we adopt a higher-density input configuration with 128 EEG channels in the 10–20 montage to improve robustness and better capture spatial patterns. Instead of relying on existing pretrained weights, we pretrain the encoder from scratch on our own curated dataset, so that the learned representations are tailored to our specific preprocessing and data distribution, which in turn leads to better transfer and performance in stage 2.
> >
> > **Q4: Regarding dimensional consistency in the EEG encoder.** Dimensional consistency is unclear: implies ( being the number of temporal patches), yet (lines 204–206). How is the channel dimension aggregated? (It seems is used? Can you elaborate on what exactly this is?) Also how does it handle varying number of channels during training?
> >
> > **A4:** Thank you for your comments. To address your first point, the channel dimension is not aggregated at the input stage of our CALAR module. This step is applied only after the main Transformer encoder (in our EEG Encoder), where we pool the final output sequence to produce the single-vector embedding, $E_{proj}$ .
> >
> > To address your second question on handing varying channel input, our architecture achieves this by using the spatial encodings $\mathcal{E}$.  Specifically, we segment the signal from all available $C$ channels into $K$ temporal patches each, creating a variable-length sequence of $N = C \times K$ total patches. As the Transformer architecture is designed for variable-length inputs, we batch these sequences by padding them to a uniform length and using an attention mask to ignore the padding tokens.  The model handles this variability because $\mathcal{E}$ is a learnable lookup table that provides a unique vector corresponding to each channel's specific 10-20 system identity (e.g., 'Cz', 'Pz').  This spatial embedding is added to every patch, providing the Transformer with the necessary context about which channel each patch came from and enabling it to handle the "heterogeneous channel configurations" listed in Table 1.
> >
> > To address your two concerns, we have revised Section 3.2 to make this prcocess explicit.
> >
> > **Q5: Regarding missing implementation details.** Important training configurations are missing (e.g., number of layers, embedding dimension, optimizer hyperparameters, fine-tuning LR, rank for CALRA, etc.).
> >
> > **A5:** Thanks for your suggestion. It is true that these key details were missing from the main paper.
> > We have summarized the most important training configurations for our MindMix model below.
> >
> > | Parameter Category | Hyperparameter | Value |
> > | --- | --- | --- |
> > | EEG Encoder | Transformer Layers | 12 |
> > |  | Embedding Dimension | 200 |
> > |  | Attention Head | 10 |
> > |  | Feed-forward Dimension | 800 |
> > | Patch Encoder | Tpye | 3-layer 1D CNN |
> > |  | Patch Dimension | 200  |
> > |  | Output Channels | 8 |
> > | CALRA Module | Input/Output Dimension | 256 |
> > |  | Low-Rank Dimension | 128 |
> > |  | Attention HJeads | 4 |
> > |  | FNN Hidden Dimension | 512 |
> > | Optimizer | Tpye | AdamW |
> > |  | Fine-tuning LR | 1e-5 |
> > |  | Weight Decay | 0.01 |
> > |  | Adam Betas | (0.9,0.95) |
> > |  | Warmup Eppchs | 3 |
> >
> > The full implementation contains numerous other hyperparameters which are too extensive to list exhaustively here. For complete reproducibility, all model definitions and configuration files are available in our code repository. We have also added this comprehensive summary to the Appendix A.3 of our revised manuscript.

---

> ### Author Response · Authors · 2025-11-22
>
> **Q6: Regarding the novelty of the CALRA module.** I am not an expert in CLIP style training, but it seems the CALRA module resembles prior cross-attention alignment mechanisms used in multimodal CLIP-style works (e.g., *CARZero: Cross-Attention Alignment for Radiology Zero-Shot Classification*, *Multi-Granularity Cross-Modal Alignment*). Please differentiate CALRA from these or acknowledge conceptual overlap, and add a related works for CLIP modality alignment especially using CALRA like modules.
>
> **A6:** Thank you for this insightful comment and for pointing us to the relevant work on CARZero [R4] and MGCA [R5]. We will clarify that CALRA's architectural purpose differs significantly. Versus MGCA: MGCA's CTA module focuses on learning a soft matching between fine-grained visual and text tokens to drive alignment at a local level. Versus CARZero: CARZero employs cross-attention to replace the standard similarity function, using it to compute a "Similarity Representation (SimR)" which is then projected to a final score. Our CALRA module is neither a local-level aligner nor a similarity function. Instead, CALRA is designed as a feature refinement module. It performs deep, symmetric interaction on the global EEG and Audio representations to produce two new, enhanced global embeddings ($E_{aligned}$ and $A_{aligned}$). These refined embeddings, not a score, are the final outputs that are aligned using the standard InfoNCE loss. In our revised manuscript, we have updated Section 3.3 to include this important distinction with these CLIP related works, emphasizing our "refine-then-contrast" strategy and better motivating our design choices relative to this prior art.
>
> **Q7: Regarding the justification and motivation for CALRA.** Beyond performance gains, please explain in more detail *why* CALRA works with proper references to previous CLIP works if possible—e.g., what failure modes of simpler co-attention or projection methods it solves. In its current form, the architecture seems to have come out of nowhere with no context explained.
>
> **A7:** Thanks for pointing this out. Indeed, the motivation behind the design of CALRA is not sufficiently detailed in current manuscript. We have revised the manuscript to make our reasoning explicit, starting from the core challenge we aim to solve.
>
> The primary motivation for CALRA is to achieve a deep, fine-grained, and robust semantic alignment capable of handling the unique challenges of auditory decoding. This task is especially difficult due to two factors: (1) the inherently low signal-to-noise ratio of EEG and the highly complex, non-linear mapping to acoustic features, and (2) the heterogeneous nature of the stimuli, as neural responses vary significantly for different audio types (e.g., speech vs. music), demanding more than a single, uniform mapping. This specific gap cannot be filled by simpler paradigms: "shallow projection" (like standard CLIP) is too weak for this complex mapping, while "simple fusion" (like co-attention) breaks the two-stream architecture required by contrastive learning.
>
> CALRA was therefore designed to fill this gap. It achieves deep interaction (which 'shallow projection' lacks) while preserving modality identity (which 'simple fusion' breaks). Its components are synergistic, each solving a specific, observed failure mode rooted in our task: 1) The Type-specific Aligner  addresses the heterogeneity of auditory stimuli by routing inputs through specialized, type-dependent transformations to model distinct brain responses. 2) The Cross-Attention  enables deep, fine-grained interaction between EEG neural patterns and audio features without collapsing the modality-specific representations. 3)The Shared Low-Rank Alignment  enforces a shared semantic bottleneck to promote stable alignment between noisy EEG and complex audio, reducing overfitting and improving generalization across subjects. Together, these components form an architecture specifically designed to refine representations for this difficult, multi-faceted alignment task.
>
> We have updated the introduction of Section 3.3 to explicitly frame CALRA's motivation this way, appropriately citing previous CLIP works to illustrate the defects of these paradigms. We hope this revision fully clarifies our motivation and resolves your concern.

---

> ### Author Response · Authors · 2025-11-22
>
> **Q8(a): Regarding presentation (Eq. 1 definition).** Define $\mathcal{T}$ and $\mathcal{E}$ in Eq. (1) explicitly (e.g., are they positional or channel embeddings?).
>
> **A8(a):** Thanks for pointing this out. Specifically,  $\mathcal{T}$ represents the temporal (positional) embedding, which is added to each patch to indicate its relative position.  On the other hand, $\mathcal{E}$ represents the spatial (channel) embedding, which is added to each patch to indicate its channel identity (e.g., 'Cz', 'Pz').  We have explicitly added these definitions to Section 3.2.
>
> **Q8(b): Regarding presentation (Eq. 10 definition).** Clarify what $E_i$ represents in Eq. (10) ($E_{aligned}$ or $E_{proj}$?).
>
> **A8(b):** Thanks for your comment. In Eq. (10) the $E_i$ is refers to $E_{aligned,i}$ and $A_i$ is refers to $A_{aligned,i}$, where $i$ represents the $i$-th sample in a training batch.  We have updated the equation's subscript to  $E_{aligned,i}$ and $A_{aligned,i}$ to avoid misleading.
>
> **Q8(c): Regarding presentation (Results).** The results in Table 2 and Figure 4 are scattered. They should be combined for easier comparison.
>
> **A8(c):** Thanks for your suggestion. We totally agree with you. In the revised manuscript, we have combined the results from Table 2 and Figure 4 into a single table for easier comparison.
>
> **Q8(d): Regarding presentation (Terminology).** Explain what “token-level interaction between modalities” (line 134) precisely means.
>
> **A8(d):** Thank you for this question, as it highlights a significant point of confusion caused by our descriptions. The phrase "token-level interaction" is a misuse in our text. As correctly described in Section 3.2, both our EEG and Audio encoders produce single vector embeddings ($E_{proj}$ and $A_{proj}$) via mean pooling. The CALRA module therefore performs its alignment operations on these vectors. We have removed these misleading references to token-level interaction from the revised manuscript and clarify that our innovation lies in the multi-step vector alignment pipeline (Type-specific Aligner and Shared Low-Rank Alignment). We sincerely apologize for this confusion.
>
> **Q8(e): Regarding presentation (Ablation "w/ LaBraM").** In Table 3, “Ablation on EEG Encoder”, when the author mentions “w/ LaBraM”, does this mean using the pretrained weights of LaBRaM or using their architecture only initialized from scratch?
>
> **A8(e):** Thank you for this question. "w/ LaBraM" refers to using the official pretrained LaBraM backbone with their released model weights. To clarify the process: we took this pretrained LaBraM, applied our full multimodal alignment training (using the 100+ hours of paired data and the CALRA module ), and then fine-tuned this aligned model (LaBraM + CALRA) on the downstream auditory tasks. This was done to test CALRA's effectiveness on a different SOTA backbone. We have added a clarification in the caption of Table 3  to make this process explicit.

---

> ### Author Response · Authors · 2025-11-22
>
> **Q9: Regarding the claim of sensitivity to preprocessing.** The author claims in line 351 that “… these large models are often highly sensitive to the data format and preprocessing pipelines..”, but do not provide any reference or supporting experimental evidence, and whether their model is superior.
>
> **A9:** Thanks for pointing this out. Our claim on line 351 regarding model sensitivity was based on our own internal experiments. We found that when we fine-tuned foundation models like LaBraM (pretrained on 0.1-70 Hz data ) using our 1-40 Hz preprocessing pipeline, their performance dropped significantly.  This is precisely why for the results in Table 2 , we fine-tuned each baseline (e.g., LaBraM , CBraMod ) using their offical preprocessing pipelines, as we wanted to present their fairest performance and not penalize them for this exact distribution shift. As you requested, we now provide that supporting experimental evidence. The internal preprocessing comparison is tabulated below, which clearly supports our claim:
>
> | **Model** | **Dataset** | **Accuracy (our pipeline)** | **Accuracy (their pipeline)** |
> | --- | --- | --- | --- |
> | BENDR | EEG4EMO | 0.6051 $\pm$ 0.019 | 0.6458 $\pm$ 0.015 |
> | BIOT | EEG4EMO | 0.5615 $\pm$ 0.027 | 0.6352 $\pm$ 0.023 |
> | EEGPT | EEG4EMO | 0.6527 $\pm$ 0.064 | 0.7129 $\pm$ 0.072 |
> | CBraMod | EEG4EMO | 0.6680 $\pm$ 0.083 | 0.7295 $\pm$ 0.082 |
> | LabraM | EEG4EMO | 0.6701 $\pm$ 0.085 | 0.7285 $\pm$ 0.078 |
> | **Model** | **Dataset** | **Duo Accuracy (our pipeline)** | **Duo Accuracy (their pipeline)** |
> | BENDR | MAD-EEG | 0.5844 $\pm$ 0.054 | 0.6235 $\pm$ 0.048 |
> | BIOT | MAD-EEG | 0.6042 $\pm$ 0.042 | 0.6485 $\pm$ 0.052 |
> | EEGPT | MAD-EEG | 0.7311 $\pm$ 0.074 | 0.7887 $\pm$ 0.065 |
> | CBraMod | MAD-EEG | 0.7007 $\pm$ 0.075 | 0.7582 $\pm$ 0.082 |
> | LabraM | MAD-EEG | 0.7554 $\pm$ 0.066 | 0.8011 $\pm$ 0.069 |
>
> As the table shows, this distribution shift causes a consistent performance drop across all models on both datasets when using our mismatched pipeline. On EEG4EMO, this drop ranges from 4-7 percentage points, with BIOT showing a particularly steep decline from 0.6352 to 0.5615. This trend holds for the SOTA models as well; for instance, CBraMod's accuracy falls from 0.7295 to 0.6680, and LaBraM's falls from 0.7285 to 0.6701. The same pattern is seen on MAD-EEG, where LaBraM's Duo Accuracy drops from 0.8011 to 0.7554 and EEGPT's drops from 0.7887 to 0.7311.
>
> Regarding our model's superiority, its advantage comes from being pretrained and aligned from the start using the same 1-40 Hz pipeline as the downstream tasks, thereby avoiding this mismatch. We have incorporated this experiment and corresponding analysis in Appendix A.2 of the updated paper.
>
> **Q10: Regarding the claim that the EEG-only model "often outperforms SOTA"**. The author claims in line 417 that their EEG-only counterpart often outperforms SOTA. However, when comparing Table 2 with Figure 4 we find that CBraMod is beaten by the EEG-only model only in the ESAA task. Please clarify this discrepancy.
>
> **A10**: Thanks for pointing this out. You are correct that our wording on line 417 was imprecise.  Our original claim was based on a comparison against LaBraM , which we viewed as the primary SOTA baseline in Table 2 as it obtained the top performance in 7 out of 12 metrics (compared to 5 for CBraMod).  However, we agree that the phase"often outperforms" is an overstatement and creates ambiguity. To resolve this, we have revised the sentence to state that our EEG-Only model is "highly competitive" with SOTA baselines, which is a more accurate and defensible claim.
>
> **Q11: Regarding harmonizing preprocessing pipelines.** The model uses data that was preprocessed using a bandpass filter of 1–40 Hz, whereas baselines you used such as LaBraM was pretrained on data bandpassed at 0.1–70 Hz. Were preprocessing pipelines harmonized across comparisons? (Or preprocessing set differently based on the basemodel used?) Differences here can cause distribution shifts that unfairly penalize other pretrained models.
>
> **A11:** Thanks for your question. It is true that forcing a pretrained model like LabraM (pretrained on 0.1-70Hz data) onto a new dataset filter at 1-40 Hz would cause a distribution shift and unfairly penalize its performance. This phenomenon is strongly confirmed by the supporting experimental evidence we provided in our response to Q9. Therefore, to ensure a fair and rigorous comparison, our this work adhered as closely as possible to the official preprocessing pipelines described in the original papers or code for each foundation model (e.g., 0.1-70 Hz for LaBraM and 0.3-75 Hz for CBraMod). Our 1-40 Hz pipeline was used only for our model (MindMix) and for the task-specific models (like DARNet and DBPNet) that were trained from scratch. The corresponding description regarding the preprocessing pipeline usage has been added to Appendix A.2(Data Preprocessing) in detail for clarity.

---

> ### Author Response · Authors · 2025-11-22
>
> **Q12: Regarding EEG-only model generalization.** The author claims in line 417 that their EEG-only counterpart often outperforms SOTA. It might beneficial to verify this with tasks other than auditory tasks. (More general tasks used in the previous models’ papers)
>
> **A12:** Thank you for your suggestion. This point is directly related to our (now corrected) claim on line 417. As we acknowledged in our response to Q10, our original claim of "often outperforms SOTA" was an overstatement. We have already revised this in the manuscript to the more accurate claim that our EEG-Only model is "highly competitive". While we agree that benchmarking our encoder on general non-auditory tasks (like those in the LaBraM) would be an interesting analysis, the primary focus of this paper is not to introduce a new state-of-the-art general-purpose EEG encoder.
> However, to test the generalization of our EEG encoder as you suggested, we have now benchmarked our EEG-Only model on two standard non-auditory tasks: TUAB [R5] (abnormal detection)  and BCI Competition IV-2b [R6] (motor imagery). We followed the standard evaluation protocols for both tasks and compared our model to reported SOTA baselines.
>
> | **Model** | **Dataset** | **Balanced Accuracy** | **Wighted F1** |
> | --- | --- | --- | --- |
> | BENDR | TUAB | 0.7915 $\pm$ 0.007 | 0.8522 $\pm$ 0.004 |
> | BIOT | TUAB | 0.7844 $\pm$ 0.005 | 0.8854 $\pm$ 0.003 |
> | EEGPT | TUAB | 0.8833 $\pm$ 0.002 | 0.9432 $\pm$ 0.001 |
> | LabraM | TUAB | 0.8210 $\pm$ 0.003 | 0.8979 $\pm$ 0.002 |
> | CBraMod | TUAB | 0.8289 $\pm$ 0.005 | 0.9018 $\pm$ 0.002 |
> | MindMix (Encoder-only) | TUAB | 0.8545 $\pm$ 0.004 | 0.9113 $\pm$ 0.005 |
> | **Model** | **Dataset** | **Balanced Accuracy** | **Wighted F1** |
> | BENDR | BCIC-IV-2B | 0.6806 $\pm$ 0.007 | 06801 $\pm$ 0.007 |
> | BIOT | BCIC-IV-2B | 0.5524 $\pm$ 0.010 | 0.5516 $\pm$ 0.010 |
> | EEGPT | BCIC-IV-2B | 0.6893 $\pm$ 0.009 | 0.6890 $\pm$ 0.009 |
> | LabraM | BCIC-IV-2B | 0.6610 $\pm$ 0.011 | 0.6608 $\pm$ 0.011 |
> | CBraMod | BCIC-IV-2B | 0.6910 $\pm$ 0.008 | 0.6898 $\pm$ 0.008 |
> | MindMix (Encoder-only) | BCIC-IV-2B | 0.6943 $\pm$ 0.010 | 0.6921 $\pm$ 0.010 |
>
> As we can see from this table, the performance of our EEG-only model is indeed highly competitive. On the TUAB dataset, our encoder obtained the second best performance in terms of balanced accuracy and weighted F1 score. On the BCIC-IV-2B dataset, our encoder achieved the highest performance among all listed foundation models. To address your concern, we have added the experiment results and corresponding disscussion in Appendix A.8 of our updated paper.
>
> **Q13: Regarding isolating CALRA's contribution.** Given that CBraMod still outperforms your EEG-only model (as pointed out by the ”Miscellaneous but important points” I wrote above), have you tried CLIP-style pretraining on top of CBraMod to isolate CALRA’s contribution?
>
> **A13:** Thank you for this suggestion. To clearly address your request to "isolate CALRA's contribution," we must clarify that the performance gain comes from our entire Stage 2 Multimodal Alignment Framework (which includes the 100h+ paired data, the Audio Encoder, and the CALRA module), rather than CALRA alone. We performed the requested ablation by applying this identical alignment framework to the official pretrained CBraMod backbone as shown below:
>
> | **Model Configuration** | **Emotion Acc. (EEG4EMO)** | **AAD Acc. (KUL)** |
> | --- | --- | --- |
> | MindMix | 0.8878 ± 0.045 | 0.9982 ± 0.008 |
> | w/ LaBram | 0.8588 ± 0.041 | 0.9744 ± 0.012 |
> | w/ CBraMod | 0.8642 ± 0.039 | 0.9637 ± 0.010 |
>
> We can conclude two observations from the above results:
>
> 1. Effect of Alignment Framework: CBraMod's performance jumps from 68.42% (unimodal) to 96.37% (aligned). This proves that our alignment strategy is highly effective and generalizable.
>
> 2. Effect of EEG Encoder: Even with this massive boost, the MindMix (Full Model) still significantly outperforms the aligned CBraMod (99.82% vs. 96.37%).
>
> Since both models in this comparison use the exact same alignment framework, the remaining performance gap isolates the contribution of the EEG Encoder itself. This confirms that our MindMix encoder provides a superior foundation for auditory decoding compared to generic backbones like CBraMod.

---

### Meta-Review · Area_Chair_anLT · 2026-01-05

**Summary:**

The authors propose a foundational model for EEG and audio. They first train a large EEG encoder on unlabelled data, and then pair audio and EEG data to align the two modalities using their proposed CARLA module. The authors present strong results (even testing between trials)

Reviewers had numerous concerns about the paper (particularly about the experimental protocol, and further clarifications). The authors provided a strong rebuttal which addressed most of these concerns. As a result, the final decision is to accept the paper. However, the paper requires significant changes to address all of the reviewers' concerns, and this needs to be done by the camera-ready version.

**Reviewer Concerns:**

- Reviewer FBHr's concerns were addressed by the lengthy rebuttal.
- Reviewer gpHw's concerns were addressed.
- Reviewer NCy7's concerns were addressed

**Reviewer Scores:**

- Reviewer FBHr would upgrade their score to weak accept.
- Reviewer gpHw remained at weak accept.
- Reviewer NCy7 would likely remain at weak accept.
- Reviewer bKp3 appears LLM generated. Ignored.

---

### Decision · Program_Chairs · 2026-01-26

Accept (Poster)